# Parallel Direction Method of Multipliers

**Huahua Wang , Arindam Banerjee , Zhi-Quan Luo**
University of Minnesota, Twin Cities
{huwang,banerjee}@cs.umn.edu, luozq@umn.edu

## Abstract

We consider the problem of minimizing block-separable (non-smooth) convex functions subject to linear constraints. While the Alternating Direction Method of Multipliers (ADMM) for two-block linear constraints has been intensively studied both theoretically and empirically, in spite of some preliminary work, effective generalizations of ADMM to multiple blocks is still unclear. In this paper, we propose a parallel randomized block coordinate method named Parallel Direction Method of Multipliers (PDMM) to solve optimization problems with multi-block linear constraints. At each iteration, PDMM randomly updates some blocks in parallel, behaving like parallel randomized block coordinate descent. We establish the global convergence and the iteration complexity for PDMM with constant step size. We also show that PDMM can do randomized block coordinate descent on overlapping blocks. Experimental results show that PDMM performs better than state-of-the-arts methods in two applications, robust principal component analysis and overlapping group lasso.

## 1 Introduction

In this paper, we consider the minimization of block-seperable convex functions subject to linear constraints, with a canonical form:

$$\min_{\{\mathbf{x}_j \in \mathcal{X}_j\}} f(\mathbf{x}) = \sum_{j=1}^{J} f_j(\mathbf{x}_j) , \text{ s.t. } \mathbf{A}\mathbf{x} = \sum_{j=1}^{J} \mathbf{A}_j^c \mathbf{x}_j = \mathbf{a} , \tag{1}$$

where the objective function $f(\mathbf{x})$ is a sum of $J$ block separable (nonsmooth) convex functions, $\mathbf{A}_j^c \in \mathbb{R}^{m \times n_j}$ is the $j$-th column block of $\mathbf{A} \in \mathbb{R}^{m \times n}$ where $n = \sum_j n_j$, $\mathbf{x}_j \in \mathbb{R}^{n_j \times 1}$ is the $j$-th block coordinate of $\mathbf{x}$, $\mathcal{X}_j$ is a local convex constraint of $\mathbf{x}_j$ and $\mathbf{a} \in \mathbb{R}^{m \times 1}$. The canonical form can be extended to handle linear inequalities by introducing slack variables, i.e., writing $\mathbf{A}\mathbf{x} \leq \mathbf{a}$ as $\mathbf{A}\mathbf{x} + \mathbf{z} = \mathbf{a}, \mathbf{z} \geq \mathbf{0}$.

A variety of machine learning problems can be cast into the linearly-constrained optimization problem (1) [8, 4, 24, 5, 6, 21, 11]. For example, in robust Principal Component Analysis (RPCA) [5], one attempts to recover a low rank matrix $\mathbf{L}$ and a sparse matrix $\mathbf{S}$ from an observation matrix $\mathbf{M}$, i.e., the linear constraint is $\mathbf{M} = \mathbf{L} + \mathbf{S}$. Further, in the stable version of RPCA [29], an noisy matrix $\mathbf{Z}$ is taken into consideration, and the linear constraint has three blocks, i.e., $\mathbf{M} = \mathbf{L} + \mathbf{S} + \mathbf{Z}$. Problem (1) can also include composite minimization problems which solve a sum of a loss function and a set of nonsmooth regularization functions. Due to the increasing interest in structural sparsity [1], composite regularizers have become widely used, e.g., overlapping group lasso [28]. As the blocks are overlapping in this class of problems, it is difficult to apply block coordinate descent methods for large scale problems [16, 18] which assume block-separable. By simply splitting blocks and introducing equality constraints, the composite minimization problem can also formulated as (1) [2].

A classical approach to solving (1) is to relax the linear constraints using the (augmented) Lagrangian, i.e.,

$$L_\rho(\mathbf{x}, \mathbf{y}) = f(\mathbf{x}) + \langle \mathbf{y}, \mathbf{A}\mathbf{x} - \mathbf{a} \rangle + \frac{\rho}{2} \|\mathbf{A}\mathbf{x} - \mathbf{a}\|_2^2 , \tag{2}$$

where $\rho \geq 0$ is called the penalty parameter. We call $\mathbf{x}$ the primal variable and $\mathbf{y}$ the dual variable. (2) usually leads to primal-dual algorithms which update the primal and dual variables alternatively. While the dual update is simply dual gradient descent, the primal update is to solve a minimization problem of (2) given $\mathbf{y}$. If $\rho = 0$, the primal update can be solved in a parallel block coordinate fashion [3, 19], leading to the dual ascent method. While the dual ascent method can achieve massive parallelism, a careful choice of stepsize and some strict conditions are required for convergence, particularly when $f$ is nonsmooth. To achieve better numerical efficiency and convergence behavior compared to the dual ascent method, it is favorable to set $\rho > 0$ in the augmented Lagrangian (2) which we call the method of multipliers. However, (2) is no longer separable and solving entire augmented Lagrangian (2) exactly is computationally expensive. In [20], randomized block coordinate descent (RBCD) [16, 18] is used to solve (2) exactly, but leading to a double-loop algorithm along with the dual step. More recent results show (2) can be solved inexactly by just sweeping the coordinates once using the alternating direction method of multipliers (ADMM) [12, 2]. This paper attempts to develop a parallel randomized block coordinate variant of ADMM.

When $J = 2$, ADMM has been widely used to solve the augmented Lagragian (2) in many applications [2]. Encouraged by the success of ADMM with two blocks, ADMM has also been extended to solve the problem with multiple blocks [15, 14, 10, 17, 13, 7]. The variants of ADMM can be mainly divided into two categories. The first category considers Gauss-Seidel ADMM (GSADMM) [15, 14], which solves (2) in a cyclic block coordinate manner. In [13], a back substitution step was added so that the convergence of ADMM for multiple blocks can be proved. In some cases, it has been shown that ADMM might not converge for multiple blocks [7]. In [14], a block successive upper bound minimization method of multipliers (BSUMM) is proposed to solve the problem (1). The convergence of BSUMM is established under some fairly strict conditions: (i) certain local error bounds hold; (ii) the step size is either sufficiently small or decreasing. However, in general, Gauss-Seidel ADMM with multiple blocks is not well understood and its iteration complexity is largely open. The second category considers Jacobian variants of ADMM [26, 10, 17], which solves (2) in a parallel block coordinate fashion. In [26, 17], (1) is solved by using two-block ADMM with splitting variables (sADMM). [10] considers a proximal Jacobian ADMM (PJADMM) by adding proximal terms. A randomized block coordinate variant of ADMM named RBSUMM was proposed in [14]. However, RBSUMM can only randomly update one block. Moreover, the convergence of RBSUMM is established under the same conditions as BSUMM and its iteration complexity is unknown.

In this paper, we propose a parallel randomized block coordinate method named parallel direction method of multipliers (PDMM) which randomly picks up any number of blocks to update in parallel, behaving like randomized block coordinate descent [16, 18]. Like the dual ascent method, PDMM solves the primal update in a parallel block coordinate fashion even with the augmentation term. Moreover, PDMM inherits the merits of the method of multipliers and can solve a fairly large class of problems, including nonsmooth functions. Technically, PDMM has three aspects which make it distinct from such state-of-the-art methods. First, if block coordinates of the primal $\mathbf{x}$ is solved exactly, PDMM uses a backward step on the dual update so that the dual variable makes conservative progress. Second, the sparsity of $\mathbf{A}$ and the number of randomized blocks are taken into consideration to determine the step size of the dual update. Third, PDMM can randomly update arbitrary number of primal blocks in parallel. Moreover, we show that sADMM and PJADMM are the two extreme cases of PDMM. The connection between sADMM and PJADMM through PDMM provides better understanding of dual backward step. PDMM can also be used to solve overlapping groups in a randomized block coordinate fashion. Interestingly, the corresponding problem for RBCD [16, 18] with overlapping blocks is still an open problem. We establish the global convergence and $O(1/T)$ iteration complexity of PDMM with constant step size. We evaluate the performance of PDMM in two applications: robust principal component analysis and overlapping group lasso.

The rest of the paper is organized as follows: We introduce PDMM in Section 2, and establish convergence results in Section 3. We evaluate the performance of PDMM in Section 4 and conclude in Section 5. The technical analysis and detailed proofs are provided in the supplement.

**Notations:** Assume that $\mathbf{A} \in \mathbb{R}^{m \times n}$ is divided into $I \times J$ blocks. Let $\mathbf{A}_i^r \in \mathbb{R}^{m_i \times n}$ be the $i$-th row block of $\mathbf{A}$, $\mathbf{A}_j^c \in \mathbb{R}^{m \times n_j}$ be the $j$-th column block of $\mathbf{A}$, and $\mathbf{A}_{ij} \in \mathbb{R}^{m_i \times n_j}$ be the $ij$-th block of $\mathbf{A}$. Let $\mathbf{y}_i \in \mathbb{R}^{m_i \times 1}$ be the $i$-th block of $\mathbf{y} \in \mathbb{R}^{m \times 1}$. Let $\mathcal{N}(i)$ be a set of nonzero blocks $\mathbf{A}_{ij}$ in the

$i$-th row block $\mathbf{A}_i^r$ and $d_i = |\mathcal{N}(i)|$ be the number of nonzero blocks. Let $\tilde{K}_i = \min\{d_i, K\}$ where $K$ is the number of blocks randomly chosen by PDMM and $T$ be the number of iterations.

## 2 Parallel Direction Method of Multipliers

Consider a direct Jacobi version of ADMM which updates all blocks in parallel:

$$\mathbf{x}_j^{t+1} = \mathrm{argmin}_{\mathbf{x}_j \in \mathcal{X}_j} \, L_\rho(\mathbf{x}_j, \mathbf{x}_{k \neq j}^t, \mathbf{y}^t) \, , \tag{3}$$

$$\mathbf{y}^{t+1} = \mathbf{y}^t + \tau\rho(\mathbf{A}\mathbf{x}^{t+1} - \mathbf{a}) \, . \tag{4}$$

where $\tau$ is a shrinkage factor for the step size of the dual gradient ascent update. However, empirical results show that it is almost impossible to make the direct Jacobi updates (3)-(4) to converge even when $\tau$ is extremely small. [15, 10] also noticed that the direct Jacobi updates may not converge.

To address the problem in (3) and (4), we propose a backward step on the dual update. Moreover, instead of updating all blocks, the blocks $\mathbf{x}_j$ will be updated in a parallel randomized block coordinate fashion. We call the algorithm Parallel Direction Method of Multipliers (PDMM). PDMM first randomly select $K$ blocks denoted by set $\mathbb{J}_t$ at time $t$, then executes the following iterates:

$$\mathbf{x}_{j_t}^{t+1} = \underset{\mathbf{x}_{j_t} \in \mathcal{X}_{j_t}}{\mathrm{argmin}} \, L_\rho(\mathbf{x}_{j_t}, \mathbf{x}_{k \neq j_t}^t, \hat{\mathbf{y}}^t) + \eta_{j_t} B_{\phi_{j_t}}(\mathbf{x}_{j_t}, \mathbf{x}_{j_t}^t) \, , \, j_t \in \mathbb{J}_t, \tag{5}$$

$$\mathbf{y}_i^{t+1} = \mathbf{y}_i^t + \tau_i\rho(\mathbf{A}_i\mathbf{x}^{t+1} - \mathbf{a}_i) \, , \tag{6}$$

$$\hat{\mathbf{y}}_i^{t+1} = \mathbf{y}_i^{t+1} - \nu_i\rho(\mathbf{A}_i\mathbf{x}^{t+1} - \mathbf{a}_i) \, , \tag{7}$$

where $\tau_i > 0, 0 \leq \nu_i < 1, \eta_{j_t} \geq 0$, and $B_{\phi_{j_t}}(\mathbf{x}_{j_t}, \mathbf{x}_{j_t}^t)$ is a Bregman divergence. Note $\mathbf{x}^{t+1} = (\mathbf{x}_{\mathbb{J}_t}^{t+1}, \mathbf{x}_{k \notin \mathbb{J}_t}^t)$ in (6) and (7). (6) and (7) update all dual blocks. We show that PDMM can also do randomized dual block coordinate ascent in an extended work [25]. Let $\tilde{K}_i = \min\{d_i, K\}$. $\tau_i$ and $\nu_i$ can take the following values:

$$\tau_i = \frac{K}{\tilde{K}_i(2J - K)} \, , \nu_i = 1 - \frac{1}{\tilde{K}_i} \, . \tag{8}$$

In the $\mathbf{x}_{j_t}$-update (5), a Bregman divergence is adddded so that exact PDMM and its inexact variants can be analyzed in an unified framework [23, 11]. In particular, if $\eta_{j_t} = 0$, (5) is an exact update. If $\eta_{j_t} > 0$, by choosing a suitable Bregman divergence, (5) can be solved by various inexact updates, often yielding a closed-form for the $\mathbf{x}_{j_t}$ update (see Section 2.1).

To better understand PDMM, we discuss the following three aspects which play roles in choosing $\tau_i$ and $\nu_i$: the dual backward step (7), the sparsity of $\mathbf{A}$, and the choice of randomized blocks.

**Dual Backward Step:** We attribute the failure of the Jacobi updates (3)-(4) to the following observation in (3), which can be rewritten as:

$$\mathbf{x}_j^{t+1} = \mathrm{argmin}_{\mathbf{x}_j \in \mathcal{X}_j} \, f_j(\mathbf{x}_j) + \langle \mathbf{y}^t + \rho(\mathbf{A}\mathbf{x}^t - \mathbf{a}), \mathbf{A}_j^c \mathbf{x}_j \rangle + \frac{\rho}{2}\|\mathbf{A}_j^c(\mathbf{x}_j - \mathbf{x}_j^t)\|_2^2 \, . \tag{9}$$

In the primal $\mathbf{x}_j$ update, the quadratic penalty term implicitly adds full gradient ascent step to the dual variable, i.e., $\mathbf{y}^t + \rho(\mathbf{A}\mathbf{x}^t - \mathbf{a})$, which we call implicit dual ascent. The implicit dual ascent along with the explicit dual ascent (4) may lead to too aggressive progress on the dual variable, particularly when the number of blocks is large. Based on this observation, we introduce an intermediate variable $\hat{\mathbf{y}}^t$ to replace $\mathbf{y}^t$ in (9) so that the implicit dual ascent in (9) makes conservative progress, e.g., $\hat{\mathbf{y}}^t + \rho(\mathbf{A}\mathbf{x}^t - \mathbf{a}) = \mathbf{y}^t + (1 - \nu)\rho(\mathbf{A}\mathbf{x}^t - \mathbf{a})$, where $0 < \nu < 1$. $\hat{\mathbf{y}}^t$ is the result of a 'backward step' on the dual variable, i.e., $\hat{\mathbf{y}}^t = \mathbf{y}^t - \nu\rho(\mathbf{A}\mathbf{x}^t - \mathbf{a})$.

Moreover, one can show that $\tau$ and $\nu$ have also been implicitly used when using two-block ADMM with splitting variables (sADMM) to solve (1) [17, 26]. Section 2.2 shows sADMM is a special case of PDMM. The connection helps in understanding the role of the two parameters $\tau_i, \nu_i$ in PDMM. Interestingly, the step sizes $\tau_i$ and $\nu_i$ can be improved by considering the block sparsity of $\mathbf{A}$ and the number of random blocks $K$ to be updated.

**Sparsity of A:** Assume $\mathbf{A}$ is divided into $I \times J$ blocks. While $\mathbf{x}_j$ can be updated in parallel, the matrix multiplication $\mathbf{A}\mathbf{x}$ in the dual update (4) requires synchronization to gather messages from all block coordinates $j_t \in \mathbb{J}_t$. For updating the $i$-th block of the dual $\mathbf{y}_i$, we need $\mathbf{A}_i\mathbf{x}^{t+1} = \sum_{j_t \in \mathbb{J}_t} \mathbf{A}_{ij_t}\mathbf{x}_{j_t}^{t+1} + \sum_{k \notin \mathbb{J}_t} \mathbf{A}_{ik}\mathbf{x}_k^t$ which aggregates "messages" from all $\mathbf{x}_{j_t}$. If $\mathbf{A}_{ij_t}$ is a block of

zeros, there is no "message" from $\mathbf{x}_{j_t}$ to $\mathbf{y}_i$. More precisely, $\mathbf{A}_i\mathbf{x}^{t+1} = \sum_{j_t \in \mathbb{J}_t \cap \mathcal{N}(i)} \mathbf{A}_{ij_t}\mathbf{x}_{j_t}^{t+1} + \sum_{k \notin \mathbb{J}_t} \mathbf{A}_{ik}\mathbf{x}_k^t$ where $\mathcal{N}(i)$ denotes a set of nonzero blocks in the $i$-th row block $\mathbf{A}_i$. $\mathcal{N}(i)$ can be considered as the set of neighbors of the $i$-th dual block $\mathbf{y}_i$ and $d_i = |\mathcal{N}(i)|$ is the degree of the $i$-th dual block $\mathbf{y}_i$. If $\mathbf{A}$ is sparse, $d_i$ could be far smaller than $J$. According to (8), a low $d_i$ will lead to bigger step sizes $\tau_i$ for the dual update and smaller step sizes for the dual backward step (7). Further, as shown in Section 2.3, when using PDMM with all blocks to solve composite minimization with overlapping blocks, PDMM can use $\tau_i = 0.5$ which is much larger than $1/J$ in sADMM.

**Randomized Blocks:** The number of blocks to be randomly chosen also has the effect on $\tau_i, \nu_i$. If randomly choosing one block ($K = 1$), then $\nu_i = 0, \tau_i = \frac{1}{2J-1}$. The dual backward step (7) vanishes. As $K$ increases, $\nu_i$ increases from 0 to $1 - \frac{1}{d_i}$ and $\tau_i$ increases from $\frac{1}{2J-1}$ to $\frac{1}{d_i}$. If updating all blocks ($K = J$), $\tau_i = \frac{1}{d_i}, \nu_i = 1 - \frac{1}{d_i}$.

PDMM does not necessarily choose any $K$ combination of $J$ blocks. The $J$ blocks can be randomly partitioned into $J/K$ groups where each group has $K$ blocks. Then PDMM randomly picks some groups. A simple way is to permutate the $J$ blocks and choose $K$ blocks cyclically.

## 2.1 Inexact PDMM

If $\eta_{j_t} > 0$, there is an extra Bregman divergence term in (5), which can serve two purposes. First, choosing a suitable Bregman divergence can lead to an efficient solution for (5). Second, if $\eta_{j_t}$ is sufficiently large, the dual update can use a large step size ($\tau_i = 1$) and the backward step (7) can be removed ($\nu_i = 0$), leading to the same updates as PJADMM [10] (see Section 2.2).

Given a continuously differentiable and strictly convex function $\psi_{j_t}$, its Bregman divergence is defiend as

$$B_{\psi_{j_t}}(\mathbf{x}_{j_t}, \mathbf{x}_{j_t}^t) = \psi_{j_t}(\mathbf{x}_{j_t}) - \psi_{j_t}(\mathbf{x}_{j_t}^t) - \langle \nabla \psi_{j_t}(\mathbf{x}_{j_t}^t), \mathbf{x}_{j_t} - \mathbf{x}_{j_t}^t \rangle, \tag{10}$$

where $\nabla \psi_{j_t}$ denotes the gradient of $\psi_{j_t}$. Rearranging the terms yields

$$\psi_{j_t}(\mathbf{x}_{j_t}) - B_{\psi_{j_t}}(\mathbf{x}_{j_t}, \mathbf{x}_{j_t}^t) = \psi_{j_t}(\mathbf{x}_{j_t}^t) + \langle \nabla \psi_{j_t}(\mathbf{x}_{j_t}^t), \mathbf{x}_{j_t} - \mathbf{x}_{j_t}^t \rangle, \tag{11}$$

which is exactly the linearization of $\psi_{j_t}(\mathbf{x}_{j_t})$ at $\mathbf{x}_{j_t}^t$. Therefore, if solving (5) exactly becomes difficult due to some problematic terms, we can use the Bregman divergence to linearize these problematic terms so that (5) can be solved efficiently. More specifically, in (5), we can choose $\phi_{j_t} = \varphi_{j_t} - \frac{1}{\eta_{j_t}}\psi_{j_t}$ assuming $\psi_{j_t}$ is the problematic term. Using the linearity of Bregman divergence,

$$B_{\phi_{j_t}}(\mathbf{x}_{j_t}, \mathbf{x}_{j_t}^t) = B_{\varphi_{j_t}}(\mathbf{x}_{j_t}, \mathbf{x}_{j_t}^t) - \frac{1}{\eta_{j_t}}B_{\psi_{j_t}}(\mathbf{x}_{j_t}, \mathbf{x}_{j_t}^t). \tag{12}$$

For instance, if $f_{j_t}$ is a logistic function, solving (5) exactly requires an iterative algorithm. Setting $\psi_{j_t} = f_{j_t}, \varphi_{j_t} = \frac{1}{2}\|\cdot\|_2^2$ in (12) and plugging into (5) yield

$$\mathbf{x}_{j_t}^{t+1} = \underset{\mathbf{x}_{j_t} \in \mathcal{X}_{j_t}}{\mathrm{argmin}} \langle \nabla f_{j_t}(\mathbf{x}_{j_t}^t), \mathbf{x}_{j_t} \rangle + \langle \hat{\mathbf{y}}^t, \mathbf{A}_{j_t}\mathbf{x}_{j_t} \rangle + \frac{\rho}{2}\|\mathbf{A}_{j_t}\mathbf{x}_{j_t} + \sum_{k \neq j_t} \mathbf{A}_k\mathbf{x}_k^t - \mathbf{a}\|_2^2 + \eta_{j_t}\|\mathbf{x}_{j_t} - \mathbf{x}_{j_t}^t\|_2^2,$$

which has a closed-form solution. Similarly, if the quadratic penalty term $\frac{\rho}{2}\|\mathbf{A}_{j_t}^c\mathbf{x}_{j_t} + \sum_{k \neq j_t} \mathbf{A}_k^c\mathbf{x}_k^t - \mathbf{a}\|_2^2$ is a problematic term, we can set $\psi_{j_t}(\mathbf{x}_{j_t}) = \frac{\rho}{2}\|\mathbf{A}_{j_t}^c\mathbf{x}_{j_t}\|_2^2$, then $B_{\psi_{j_t}}(\mathbf{x}_{j_t}, \mathbf{x}_{j_t}^t) = \frac{\rho}{2}\|\mathbf{A}_{j_t}^c(\mathbf{x}_{j_t} - \mathbf{x}_{j_t}^t)\|_2^2$ can be used to linearize the quadratic penalty term.

In (12), the nonnegativeness of $B_{\phi_{j_t}}$ implies that $B_{\varphi_{j_t}} \geq \frac{1}{\eta_{j_t}}B_{\psi_{j_t}}$. This condition can be satisfied as long as $\varphi_{j_t}$ is more convex than $\psi_{j_t}$. Technically, we assume that $\varphi_{j_t}$ is $\sigma/\eta_{j_t}$-strongly convex and $\psi_{j_t}$ has Lipschitz continuous gradient with constant $\sigma$, which has been shown in [23].

## 2.2 Connections to Related Work

Consider the case when all blocks are used in PDMM. There are also two other methods which update all blocks in parallel. If solving the primal updates exactly, two-block ADMM with splitting variables (sADMM) is considered in [17, 26]. We show that sADMM is a special case of PDMM when setting $\tau_i = \frac{1}{J}$ and $\nu_i = 1 - \frac{1}{J}$ (Appendix B in [25]). If the primal updates are solved inexactly, [10] considers a proximal Jacobian ADMM (PJADMM) by adding proximal terms where

the converge rate is improved to $o(1/T)$ given the sufficiently large proximal terms. We show that PJADMM [10] is also a special case of PDMM (Appendix C in [25]). sADMM and PJADMM are two extreme cases of PDMM. The connection between sADMM and PJADMM through PDMM can provide better understanding of the three methods and the role of dual backward step. If the primal update is solved exactly which makes sufficient progress, the dual update should take small step, e.g., sADMM. On the other hand, if the primal update takes small progress by adding proximal terms, the dual update can take full gradient step, e.g. PJADMM. While sADMM is a direct derivation of ADMM, PJADMM introduces more terms and parameters.

In addition to PDMM, RBUSMM [14] can also randomly update one block. The convergence of RBSUMM requires certain local error bounds to be hold and decreasing step size. Moreover, the iteration complexity of RBSUMM is still unknown. In contast, PDMM converges at a rate of $O(1/T)$ with the constant step size.

## 2.3 Randomized Overlapping Block Coordinate Descent

Consider the composite minimization problem of a sum of a loss function $\ell(\mathbf{w})$ and composite regularizers $g_j(\mathbf{w}_j)$:

$$\min_{\mathbf{w}} \ \ell(\mathbf{w}) + \sum_{j=1}^{L} g_j(\mathbf{w}_j) \ , \tag{13}$$

which considers $L$ overlapping groups $\mathbf{w}_j \in \mathbb{R}^{b \times 1}$. Let $J = L+1, \mathbf{x}_J = \mathbf{w}$. For $1 \leq j \leq L$, denote $\mathbf{x}_j = \mathbf{w}_j$, then $\mathbf{x}_j = \mathbf{U}_j^T \mathbf{x}_J$, where $\mathbf{U}_j \in \mathbb{R}^{b \times L}$ is the columns of an identity matrix and extracts the coordinates of $\mathbf{x}_J$. Denote $\mathbf{U} = [\mathbf{U}_1, \cdots, \mathbf{U}_L] \in \mathbb{R}^{n \times (bL)}$ and $\mathbf{A} = [\mathbf{I}_{bL}, -\mathbf{U}^T]$ where $bL$ denotes $b \times L$. By letting $f_j(\mathbf{x}_j) = g_j(\mathbf{w}_j)$ and $f_J(\mathbf{x}_J) = \ell(\mathbf{w})$, (13) can be written as:

$$\min_{\mathbf{x}} \ \sum_{j=1}^{J} f_j(\mathbf{x}_j) \quad \text{s.t.} \quad \mathbf{A}\mathbf{x} = \mathbf{0}. \tag{14}$$

where $\mathbf{x} = [\mathbf{x}_1; \cdots; \mathbf{x}_L; \mathbf{x}_{L+1}] \in \mathbb{R}^{b \times J}$. (14) can be solved by PDMM in a randomized block coordinate fashion. In $\mathbf{A}$, for $b$ rows block, there are only two nonzero blocks, i.e., $d_i = 2$. Therefore, $\tau_i = \frac{K}{2(2J-K)}, \nu_i = 0.5$. In particular, if $K = J$, $\tau_i = \nu_i = 0.5$. In contrast, sADMM uses $\tau_i = 1/J \ll 0.5, \nu_i = 1 - 1/J > 0.5$ if $J$ is larger.

**Remark 1** (a) ADMM [2] can solve (14) where the equality constraint is $\mathbf{x}_j = \mathbf{U}_j^T \mathbf{x}_J$.

(b) In this setting, Gauss-Seidel ADMM (GSADMM) and BSUMM [14] are the same as ADMM. BSUMM should converge with constant stepsize $\rho$ (not necessarily sufficiently small), although the theory of BSUMM does not include this special case.

# 3 Theoretical Results

We establish the convergence results for PDMM under fairly simple assumptions:

**Assumption 1**

*(1) $f_j : \mathbb{R}^{n_j} \mapsto \mathbb{R} \cup \{+\infty\}$ are closed, proper, and convex.*

*(2) A KKT point of the Lagrangian ($\rho = 0$ in (2)) of Problem (1) exists.*

Assumption 1 is the same as that required by ADMM [2, 22]. Assume that $\{\mathbf{x}_j^* \in \mathcal{X}_j, \mathbf{y}_i^*\}$ satisfies the KKT conditions of the Lagrangian ($\rho = 0$ in (2)), i.e.,

$$- \mathbf{A}_j^T \mathbf{y}^* \in \partial f_j(\mathbf{x}_j^*) \ , \tag{15}$$

$$\mathbf{A}\mathbf{x}^* - \mathbf{a} = 0. \tag{16}$$

During iterations, (16) is satisfied if $\mathbf{A}\mathbf{x}^{t+1} = \mathbf{a}$. Let $f_j'(\mathbf{x}_j^{t+1}) \in \partial f_j(\mathbf{x}_j^{t+1})$ where $\partial f_j$ be the subdifferential of $f_j$. For $\mathbf{x}_j^* \in \mathcal{X}_j$, the optimality conditions for the $\mathbf{x}_j$ update (5) is

$$\langle f_j'(\mathbf{x}_j^{t+1}) + \mathbf{A}_j^c[\mathbf{y}^t + (1-\nu)\rho(\mathbf{A}\mathbf{x}^t - \mathbf{a}) + \mathbf{A}_j^c(\mathbf{x}_j^{t+1} - \mathbf{x}_j^t)] + \eta_j(\nabla\phi_j(\mathbf{x}_j^{t+1}) - \nabla\phi_j(\mathbf{x}_j^t)), \mathbf{x}_j^{t+1} - \mathbf{x}_j^* \rangle \leq 0 \ .$$

When $\mathbf{A}\mathbf{x}^{t+1} = \mathbf{a}$, $\mathbf{y}^{t+1} = \mathbf{y}^t$. If $\mathbf{A}_j^c(\mathbf{x}_j^{t+1} - \mathbf{x}_j^t) = 0$, then $\mathbf{A}\mathbf{x}^t - \mathbf{a} = 0$. When $\eta_j \geq 0$, further assuming $B_{\phi_j}(\mathbf{x}_j^{t+1}, \mathbf{x}_j^t) = 0$, (15) will be satisfied. Note $\mathbf{x}_j^* \in \mathcal{X}_j$ is always satisfied in (5) in

PDMM. Overall, the KKT conditions (15)-(16) are satisfied if the following optimality conditions are satisfied by the iterates:

$$\mathbf{A}\mathbf{x}^{t+1} = \mathbf{a} \,, \mathbf{A}_j^c(\mathbf{x}_j^{t+1} - \mathbf{x}_j^t) = 0 \,, \tag{17}$$

$$B_{\phi_j}(\mathbf{x}_j^{t+1}, \mathbf{x}_j^t) = 0 \,. \tag{18}$$

The above optimality conditions are sufficient for the KKT conditions. (17) are the optimality conditions for the exact PDMM. (18) is needed only when $\eta_j > 0$.

Let $\mathbf{z}_{ij} = \mathbf{A}_{ij}\mathbf{x}_j \in \mathbb{R}^{m_i \times 1}$, $\mathbf{z}_i^r = [\mathbf{z}_{i1}^T, \cdots, \mathbf{z}_{iJ}^T]^T \in \mathbb{R}^{m_i J \times 1}$ and $\mathbf{z} = [(\mathbf{z}_1^r)^T, \cdots, (\mathbf{z}_I^r)^T]^T \in \mathbb{R}^{Jm \times 1}$. Define the residual of optimality conditions (17)-(18) as

$$R(\mathbf{x}^{t+1}) = \frac{\rho}{2}\|\mathbf{z}^{t+1} - \mathbf{z}^t\|_{\mathbf{P}_t}^2 + \frac{\rho}{2}\sum_{i=1}^{I}\beta_i\|\mathbf{A}_i^r\mathbf{x}^{t+1} - \mathbf{a}_i\|_2^2 + \sum_{j=1}^{J}\eta_j B_{\phi_j}(\mathbf{x}_j^{t+1}, \mathbf{x}_j^t) \,. \tag{19}$$

where $\mathbf{P}_t$ is some positive semi-definite matrix and $\beta_i = \frac{K}{J\tilde{K}_i}$. If $R(\mathbf{x}^{t+1}) \to 0$, (17)-(18) will be satisfied and thus PDMM converges to the KKT point $\{\mathbf{x}^*, \mathbf{y}^*\}$. Define the current iterate $\mathbf{v}^t = (\mathbf{x}_j^t, \mathbf{y}_i^t)$ and $h(\mathbf{v}^*, \mathbf{v}^t)$ as a distance from $\mathbf{v}^t$ to a KKT point $\mathbf{v}^* = (\mathbf{x}_j^* \in \mathcal{X}_j, \mathbf{y}_i^*)$:

$$h(\mathbf{v}^*, \mathbf{v}^t) = \frac{K}{J}\sum_{i=1}^{I}\frac{1}{2\tau_i\rho}\|\mathbf{y}_i^* - \mathbf{y}_i^{t-1}\|_2^2 + \tilde{\mathcal{L}}_\rho(\mathbf{x}^t, \mathbf{y}^t) + \frac{\rho}{2}\|\mathbf{z}^* - \mathbf{z}^t\|_{\mathbf{Q}}^2 + \sum_{j=1}^{J}\eta_j B_{\phi_j}(\mathbf{x}_j^*, \mathbf{x}_j^t) \,, \tag{20}$$

where $\mathbf{Q}$ is a positive semi-definite matrix and $\tilde{\mathcal{L}}_\rho(\mathbf{x}^t, \mathbf{y}^t)$ with $\gamma_i = \frac{2(J-K)}{\tilde{K}_i(2J-K)} + \frac{1}{d_i} - \frac{K}{J\tilde{K}_i}$ is

$$\tilde{\mathcal{L}}_\rho(\mathbf{x}^t, \mathbf{y}^t) = f(\mathbf{x}^t) - f(\mathbf{x}^*) + \sum_{i=1}^{I}\left\{\langle\mathbf{y}_i^t, \mathbf{A}_i^r\mathbf{x}^t - \mathbf{a}_i\rangle + \frac{(\gamma_i - \tau_i)\rho}{2}\|\mathbf{A}_i^r\mathbf{x}^t - \mathbf{a}_i\|_2^2\right\} \,. \tag{21}$$

The following Lemma shows that $h(\mathbf{v}^*, \mathbf{v}^t) \geq 0$.

**Lemma 1** *Let $\mathbf{v}^t = (\mathbf{x}_j^t, \mathbf{y}_i^t)$ be generated by PDMM (5)-(7) and $h(\mathbf{v}^*, \mathbf{v}^t)$ be defined in (20). Setting $\nu_i = 1 - \frac{1}{\tilde{K}_i}$ and $\tau_i = \frac{K}{\tilde{K}_i(2J-K)}$, we have*

$$h(\mathbf{v}^*, \mathbf{v}^t) \geq \frac{\rho}{2}\sum_{i=1}^{I}\zeta_i\|\mathbf{A}_i^r\mathbf{x}^t - \mathbf{a}_i\|_2^2 + \frac{\rho}{2}\|\mathbf{z}^* - \mathbf{z}^t\|_{\mathbf{Q}}^2 + \sum_{j=1}^{J}\eta_j B_{\phi_j}(\mathbf{x}_j^*, \mathbf{x}_j^t) \geq 0 \,. \tag{22}$$

*where $\zeta_i = \frac{J-K}{\tilde{K}_i(2J-K)} + \frac{1}{d_i} - \frac{K}{J\tilde{K}_i} \geq 0$. Moreover, if $h(\mathbf{v}^*, \mathbf{v}^t) = 0$, then $\mathbf{A}_i^r\mathbf{x}^t = \mathbf{a}_i, \mathbf{z}^t = \mathbf{z}^*$ and $B_{\phi_j}(\mathbf{x}_j^*, \mathbf{x}_j^t) = 0$. Thus, (15)-(16) are satisfied.*

In PDMM, $\mathbf{y}^{t+1}$ depends on $\mathbf{x}^{t+1}$, which in turn depends on $\mathbb{J}_t$. $\mathbf{x}^t$ and $\mathbf{y}^t$ are independent of $\mathbb{J}_t$. $\mathbf{x}^t$ depends on the observed realizations of the random variable $\xi_{t-1} = \{\mathbb{J}_1, \cdots, \mathbb{J}_{t-1}\}$. The following theorem shows that $h(\mathbf{v}^*, \mathbf{v}^t)$ decreases monotonically and thus establishes the global convergence of PDMM.

**Theorem 1** *(Global Convergence) Let $\mathbf{v}^t = (\mathbf{x}_j^t, \mathbf{y}_i^t)$ be generated by PDMM (5)-(7) and $\mathbf{v}^* = (\mathbf{x}_j^* \in \mathcal{X}_j, \mathbf{y}_i^*)$ be a KKT point satisfying (15)-(16). Setting $\nu_i = 1 - \frac{1}{\tilde{K}_i}$ and $\tau_i = \frac{K}{\tilde{K}_i(2J-K)}$, we have*

$$0 \leq \mathbb{E}_{\xi_t}h(\mathbf{v}^*, \mathbf{v}^{t+1}) \leq \mathbb{E}_{\xi_{t-1}}h(\mathbf{v}^*, \mathbf{v}^t) \,, \quad \mathbb{E}_{\xi_t}R(\mathbf{x}^{t+1}) \to 0 \,. \tag{23}$$

The following theorem establishes the iteration complexity of PDMM in an ergodic sense.

**Theorem 2** *(Iteration Complexity) Let $(\mathbf{x}_j^t, \mathbf{y}_i^t)$ be generated by PDMM (5)-(7). Let $\bar{\mathbf{x}}^T = \sum_{t=1}^{T}\mathbf{x}^t$. Setting $\nu_i = 1 - \frac{1}{\tilde{K}_i}$ and $\tau_i = \frac{K}{\tilde{K}_i(2J-K)}$, we have*

$$\mathbb{E}f(\bar{\mathbf{x}}^T) - f(\mathbf{x}^*) \leq \frac{\frac{J}{K}\left\{\sum_{i=1}^{I}\frac{1}{2\beta_i\rho}\|\mathbf{y}_i^*\|_2^2 + \tilde{\mathcal{L}}_\rho(\mathbf{x}^1, \mathbf{y}^1) + \frac{\rho}{2}\|\mathbf{z}^* - \mathbf{z}^1\|_{\mathbf{Q}}^2 + \sum_{j=1}^{J}\eta_j B_{\phi_j}(\mathbf{x}_j^*, \mathbf{x}_j^1)\right\}}{T} \,,$$

$$\mathbb{E}\sum_{i=1}^{I}\beta_i\|\mathbf{A}_i^r\bar{\mathbf{x}}^T - \mathbf{a}_i\|_2^2 \leq \frac{\frac{2}{\rho}h(\mathbf{v}^*, \mathbf{v}^0)}{T} \,.$$

*where $\beta_i = \frac{K}{J\tilde{K}_i}$, $\mathbf{Q}$ is a positive semi-definite matrix, and the expectation is over $\mathbb{J}_t$.*

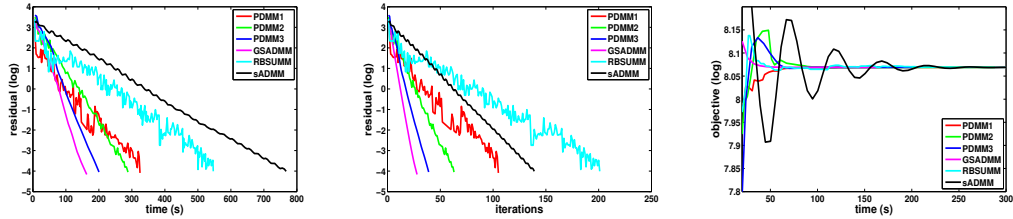

Figure 1: Comparison of the convergence of PDMM with ADMM methods in RPCA.

Table 1: The best results of PDMM with tuning parameters $\tau_i, \nu_i$ in RPCA.

|  | time (s) | iteration | residual($\times 10^{-5}$) | objective (log) |
|---|---|---|---|---|
| PDMM1 | 118.83 | 40 | 3.60 | 8.07 |
| PDMM2 | 137.46 | 34 | 5.51 | 8.07 |
| PDMM3 | 147.82 | 31 | 6.54 | 8.07 |
| GSADMM | 163.09 | 28 | 6.84 | 8.07 |
| RBSUMM | 206.96 | 141 | 8.55 | 8.07 |
| sADMM[1] | 731.51 | 139 | 9.73 | 8.07 |

**Remark 2** PDMM converges at the same rate as ADMM and its variants. In Theorem 2, PDMM can achieve the fastest convergence by setting $J = K = 1, \tau_i = 1, \nu_i = 0$, i.e., the entire matrix $\mathbf{A}$ is considered as a single block, indicating PDMM reduces to the method of multipliers. In this case, however, the resulting subproblem may be difficult to solve, as discussed in Section 1. Therefore, the number of blocks in PDMM depends on the trade-off between the number of subproblems and how efficiently each subproblem can be solved.

## 4 Experimental Results

In this section, we evaluate the performance of PDMM in solving robust principal component analysis (RPCA) and overlapping group lasso [28]. We compared PDMM with ADMM [2] or GSADMM (no theory guarantee), sADMM [17, 26], and RBSUMM [14]. Note GSADMM includes BSUMM [14]. All experiments are implemented in Matlab and run sequentially. We run the experiments 10 times and report the average results. The stopping criterion is either when the residual is smaller than $10^{-4}$ or when the number of iterations exceeds 2000.

**RPCA:** RPCA is used to obtain a low rank and sparse decomposition of a given matrix $\mathbf{A}$ corrupted by noise [5, 17]:

$$\min \frac{1}{2}\|\mathbf{X}_1\|_F^2 + \gamma_2\|\mathbf{X}_2\|_1 + \gamma_3\|\mathbf{X}_3\|_* \quad s.t. \quad \mathbf{A} = \mathbf{X}_1 + \mathbf{X}_2 + \mathbf{X}_3 . \quad (24)$$

where $\mathbf{A} \in \mathbb{R}^{m \times n}$, $\mathbf{X}_1$ is a noise matrix, $\mathbf{X}_2$ is a sparse matrix and $\mathbf{X}_3$ is a low rank matrix. $\mathbf{A} = \mathbf{L} + \mathbf{S} + \mathbf{V}$ is generated in the same way as [17][1]. In this experiment, $m = 1000, n = 5000$ and the rank is 100. The number appended to PDMM denotes the number of blocks ($K$) to be chosen in PDMM, e.g., PDMM1 randomly updates one block.

Figure 1 compares the convegence results of PDMM with ADMM methods. In PDMM, $\rho = 1$ and $\tau_i, \nu_i$ are chosen according to (8), i.e., $(\tau_i, \nu_i) = \{(\frac{1}{5}, 0), (\frac{1}{4}, \frac{1}{2}), (\frac{1}{3}, \frac{1}{3})\}$ for PDMM1, PDMM2 and PDMM3 respectively. We choose the 'best'results for GSADMM ($\rho = 1$) and RBSUMM ($\rho = 1, \alpha = \rho\frac{11}{\sqrt{t+10}}$) and sADMM ($\rho = 1$). PDMMs perform better than RBSUMM and sADMM. Note the public available code of sADMM[1] does not have dual update, i.e., $\tau_i = 0$. sADMM should be the same as PDMM3 if $\tau_i = \frac{1}{3}$. Since $\tau_i = 0$, sADMM is the slowest algorithm. Without tuning the parameters of PDMM, GSADMM converges faster than PDMM. Note PDMM can run in parallel but GSADMM only runs sequentially. PDMM3 is faster than two randomized version of PDMM since the costs of extra iterations in PDMM1 and PDMM2 have surpassed the savings at each iteration. For the two randomized one block coordinate methods, PDMM1 converges faster than RBSUMM in terms of both the number of iterations and runtime.

**The effect of $\tau_i, \nu_i$:** We tuned the parameter $\tau_i, \nu_i$ in PDMMs. Three randomized methods (RBSUMM, PDMM1 and PDMM2) choose the blocks cyclically instead of randomly. Table 1 compares the 'best'results of PDMM with other ADMM methods. In PDMM, $(\tau_i, \nu_i) =$

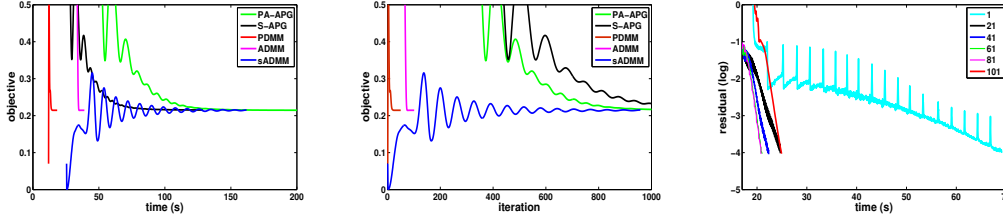

Figure 2: Comparison of convergence of PDMM and other methods in overlapping group Lasso.

$\{(\frac{1}{2}, 0), (\frac{1}{3}, \frac{1}{2}), (\frac{1}{2}, \frac{1}{2})\}$. GSADMM converges with the smallest number of iterations, but PDMMs can converge faster than GSADMM in terms of runtime. The computation per iteration in GSADMM is slightly higher than PDMM3 because GSADMM updates the sum $\mathbf{X}_1 + \mathbf{X}_2 + \mathbf{X}_3$ but PDMM3 can reuse the sum. Therefore, if the numbers of iterations of the two methods are close, PDMM3 can be faster than GSADMM. PDMM1 and PDMM2 can be faster than PDMM3. By simply updating one block, PDMM1 is the fastest algorithm and achieves the lowest residual.

**Overlapping Group Lasso:** We consider solving the overlapping group lasso problem [28]:

$$\min_{\mathbf{w}} \ \frac{1}{2L\lambda}\|\mathbf{A}\mathbf{w} - \mathbf{b}\|_2^2 + \sum_{g \in \mathcal{G}} d_g \|\mathbf{w}_g\|_2 \ . \tag{25}$$

where $\mathbf{A} \in \mathbb{R}^{m \times n}, \mathbf{w} \in \mathbb{R}^{n \times 1}$ and $\mathbf{w}_g \in \mathbb{R}^{b \times 1}$ is the vector of overlapping group indexed by $g$. $d_g$ is some positive weight of group $g \in \mathcal{G}$. As shown in Section 2.3, (25) can be rewritten as the form (14). The data is generated in a same way as [27, 9]: the elements of $\mathbf{A}$ are sampled from normal distribution, $b = \mathbf{A}x + \epsilon$ with noise $\epsilon$ sampled from normal distribution, and $\mathbf{x}_j = (-1)^j \exp(-(j-1)/100)$. In this experiment, $m = 5000$, the number of groups is $L = 100$, and $d_g = \frac{1}{L}, \lambda = \frac{L}{5}$ in (25). The size of each group is 100 and the overlap is 10. The total number of blocks in PDMM and sADMM is $J = 101$. $\tau_i, \nu_i$ in PDMM are computed according to (8).

In Figure 2, the first two figures plot the convergence of objective in terms of the number of iterations and time. PDMM uses all 101 blocks and is the fastest algorithm. ADMM is the same as GSADMM in this problem, but is slower than PDMM. Since sADMM does not consider the sparsity, it uses $\tau_i = \frac{1}{J+1}, \nu_i = 1 - \frac{1}{J+1}$, leading to slow convergence. The two accelerated methods, PA-APG [27] and S-APG [9], are slower than PDMM and ADMM.

**The effect of $K$:** The third figure shows PDMM with different number of blocks $K$. Although the complexity of each iteration is the lowest when $K = 1$, PDMM takes much more iterations than other cases and thus takes the longest time. As $K$ increases, PDMM converges faster and faster. When $K = 20$, the runtime is already same as using all blocks. When $K > 21$, PDMM takes less time to converge than using all blocks. The runtime of PDMM decreases as $K$ increases from 21 to 61. However, the speedup from 61 to 81 is negligable. We tried different set of parameters for RBSUMM $\rho \frac{i^2+1}{i+t} (0 \le i \le 5, \rho = 0.01, 0.1, 1)$ or sufficiently small step size, but could not see the convergence of the objective within 5000 iterations. Therefore, the results are not included here.

## 5 Conclusions

We proposed a randomized block coordinate variant of ADMM named Parallel Direction Method of Multipliers (PDMM) to solve the class of problem of minimizing block-separable convex functions subject to linear constraints. PDMM considers the sparsity and the number of blocks to be updated when setting the step size. We show two existing Jacobian ADMM methods are special cases of PDMM. We also use PDMM to solve overlapping block problems. The global convergence and the iteration complexity are established with constant step size. Experiments on robust PCA and overlapping group lasso show that PDMM is faster than existing methods.

## Acknowledgment

H. W. and A. B. acknowledge the support of NSF via IIS-1447566, IIS-1422557, CCF-1451986, CNS-1314560, IIS-0953274, IIS-1029711, IIS-0916750, and NASA grant NNX12AQ39A. H. W. acknowledges the support of DDF (2013-2014) from the University of Minnesota. A.B. acknowledges support from IBM and Yahoo. Z.Q. Luo is supported in part by the US AFOSR via grant number FA9550-12-1-0340 and the National Science Foundation via grant number DMS-1015346.

## Footnotes

[1]http://www.stanford.edu/ boyd/papers/prox_algs/matrix_decomp.html

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
