[Supplementary Material · supplement.pdf]

# Supplement to Parallel Direction Method of Multipliers

## 1 Convergence

We consider the minimization of block-seperable convex functions subject to linear constraints:

$$\min_{\{\mathbf{x}_j \in \mathcal{X}_j\}} f(\mathbf{x}) = \sum_{j=1}^{J} f_j(\mathbf{x}_j) \text{ , s.t. } \mathbf{A}\mathbf{x} = \sum_{j=1}^{J} \mathbf{A}_j^c \mathbf{x}_j = \mathbf{a} . \tag{1}$$

The (augmented) Lagrangian of (1) is

$$L_\rho(\mathbf{x}, \mathbf{y}) = f(\mathbf{x}) + \langle \mathbf{y}, \mathbf{A}\mathbf{x} - \mathbf{a} \rangle + \frac{\rho}{2} \|\mathbf{A}\mathbf{x} - \mathbf{a}\|_2^2 , \tag{2}$$

where $\rho \geq 0$ is the penalty parameter. PDMM has the following iterates:

$$\mathbf{x}_{j_t}^{t+1} = \operatorname*{argmin}_{\mathbf{x}_{j_t} \in \mathcal{X}_{j_t}} L_\rho(\mathbf{x}_{j_t}, \mathbf{x}_{k \neq j_t}^t, \hat{\mathbf{y}}^t) + \eta_{j_t} B_{\phi_{j_t}}(\mathbf{x}_{j_t}, \mathbf{x}_{j_t}^t) , \ j_t \in \mathbb{I}_t, \tag{3}$$

$$\mathbf{y}_i^{t+1} = \mathbf{y}_i^t + \tau_i \rho(\mathbf{A}_i^r \mathbf{x}^{t+1} - \mathbf{a}_i) , \tag{4}$$

$$\hat{\mathbf{y}}_i^{t+1} = \mathbf{y}_i^{t+1} - \nu_i \rho(\mathbf{A}_i^r \mathbf{x}^{t+1} - \mathbf{a}_i) , \tag{5}$$

### 1.1 Technical Preliminaries

We first define some notations will be used specifically in this section. Let $\mathbf{z}_{ij} = \mathbf{A}_{ij}\mathbf{x}_j \in \mathbb{R}^{m_i \times 1}$, $\mathbf{z}_i^r = [\mathbf{z}_{i1}^T, \cdots, \mathbf{z}_{iJ}^T]^T \in \mathbb{R}^{m_i J \times 1}$ and $\mathbf{z} = [(\mathbf{z}_1^r)^T, \cdots, (\mathbf{z}_I^r)^T]^T \in \mathbb{R}^{Jm \times 1}$. Let $\mathbf{W}_i \in \mathbb{R}^{Jm_i \times m_i}$ be a column vector of $\mathbf{W}_{ij} \in \mathbb{R}^{m_i \times m_i}$ where

$$\mathbf{W}_{ij} = \begin{cases} \mathbf{I}_{m_i} , & \text{if } \mathbf{A}_{ij} \neq \mathbf{0} , \\ \mathbf{0} & \text{otherwise} . \end{cases} \tag{6}$$

Define $\mathbf{Q} \in \mathbb{R}^{Jm \times Jm}$ as a diagonal matrix of $\mathbf{Q}_i \in \mathbb{R}^{Jm_i \times Jm_i}$ and

$$\mathbf{Q} = \operatorname{diag}([\mathbf{Q}_1, \cdots, \mathbf{Q}_I]) , \mathbf{Q}_i = \operatorname{diag}(\mathbf{W}_i) - \frac{1}{d_i} \mathbf{W}_i \mathbf{W}_i^T . \tag{7}$$

Therefore, for an optimal solution $\mathbf{x}^*$ satisfying $\mathbf{A}\mathbf{x}^* = \mathbf{a}$, we have

$$\|\mathbf{z}^t - \mathbf{z}^*\|_{\mathbf{Q}}^2 = \sum_{i=1}^{I} \|\mathbf{z}_i^t - \mathbf{z}_i^*\|_{\mathbf{Q}_i}^2 = \sum_{i=1}^{I} \|\mathbf{z}_i^t - \mathbf{z}_i^*\|_{\operatorname{diag}(\mathbf{w}_i) - \frac{1}{d_i}\mathbf{w}_i \mathbf{w}_i^T}^2$$

$$= \sum_{i=1}^{I} \left[ \sum_{j \in \mathcal{N}(i)} \|\mathbf{z}_{ij}^t - \mathbf{z}_{ij}^*\|_2^2 - \frac{1}{d_i} \|\mathbf{w}_i^T(\mathbf{z}_i^t - \mathbf{z}_i^*)\|_2^2 \right]$$

$$= \sum_{i=1}^{I} \left[ \|\mathbf{z}_i^t - \mathbf{z}_i^*\|_2^2 - \frac{1}{d_i} \|\mathbf{A}_i^r \mathbf{x}^t - \mathbf{a}_i\|_2^2 \right] , \tag{8}$$

where the last equality uses $\mathbf{w}_i^T \mathbf{z}_i^* = \mathbf{A}_i^r \mathbf{x}^* = \mathbf{a}_i$.

In the following lemma, we prove that $\mathbf{Q}_i$ is a positive semi-definite matrix. Thus, $\mathbf{Q}$ is also positive semi-definite.

**Lemma 1** $\mathbf{Q}_i$ *is positive semi-definite.*

*Proof:*   As $\mathbf{W}_{ij}$ is either an identity matrix or a zero matrix, $\mathbf{W}_i$ has $d_i$ nonzero entries. Removing the zero entries from $\mathbf{W}_i$, we have $\tilde{\mathbf{W}}_i$ which only has $d_i$ nonzero entries. Then,

$$\tilde{\mathbf{W}}_i = \begin{bmatrix} \mathbf{I}_{m_i} \\ \vdots \\ \mathbf{I}_{m_i} \end{bmatrix} , \mathrm{diag}(\tilde{\mathbf{W}}_i) = \begin{bmatrix} \mathbf{I}_{m_i} \\ & \ddots \\ & & \mathbf{I}_{m_i} \end{bmatrix} , \tag{9}$$

$\mathrm{diag}(\mathbf{W}_i)$ is an identity matrix. Define $\tilde{\mathbf{Q}}_i = \mathrm{diag}(\tilde{\mathbf{W}}_i) - \frac{1}{d_i} \tilde{\mathbf{W}}_i \tilde{\mathbf{W}}_i^T$. If $\tilde{\mathbf{Q}}_i$ is positive semi-definite, $\mathbf{Q}_i$ is positive semi-definite.

Denote $\lambda_{\tilde{\mathbf{W}}_i}^{\max}$ as the largest eigenvalue of $\tilde{\mathbf{W}}_i \tilde{\mathbf{W}}_i^T$, which is equivalent to the largest eigenvalue of $\tilde{\mathbf{W}}_i^T \tilde{\mathbf{W}}_i$. Since $\tilde{\mathbf{W}}_i^T \tilde{\mathbf{W}}_i = d_i \mathbf{I}_{m_i}$, then $\lambda_{\tilde{\mathbf{W}}_i}^{\max} = d_i$. Then, for any $\mathbf{v}$,

$$\|\mathbf{v}\|_{\tilde{\mathbf{W}}_i \tilde{\mathbf{W}}_i^T}^2 \le \lambda_{\tilde{\mathbf{W}}_i}^{\max} \|\mathbf{v}\|_2^2 = d_i \|\mathbf{v}\|_2^2 . \tag{10}$$

Thus,

$$\|\mathbf{v}\|_{\mathbf{Q}_i}^2 = \|\mathbf{v}\|_{\mathrm{diag}(\tilde{\mathbf{W}}_i) - \frac{1}{d_i} \tilde{\mathbf{W}}_i \tilde{\mathbf{W}}_i^T}^2 = \|\mathbf{v}\|_2^2 - \frac{1}{d_i} \|\mathbf{v}\|_{\tilde{\mathbf{W}}_i \tilde{\mathbf{W}}_i^T}^2 \ge 0 , \tag{11}$$

which completes the proof.                                                                ∎

Let $\mathbf{W}_i^t \in \mathbb{R}^{Jm_i \times m_i}$ be a column vector of $\mathbf{W}_{ij_t} \in \mathbb{R}^{m_i \times m_i}$ where

$$\mathbf{W}_{ij_t} = \begin{cases} \mathbf{I}_{m_i} , & \text{if } \mathbf{A}_{ij_t} \ne \mathbf{0} \text{ and } j_t \in \mathbb{I}_t , \\ \mathbf{0} & \text{otherwise} . \end{cases} \tag{12}$$

Define $\mathbf{P}_t \in \mathbb{R}^{Jm \times Jm}$ as a diagonal matrix of $\mathbf{P}_i^t \in \mathbb{R}^{Jm_i \times Jm_i}$ and

$$\mathbf{P}_t = \mathrm{diag}[\mathbf{P}_1^t, \cdots, \mathbf{P}_I^t] , \mathbf{P}_i^t = \mathrm{diag}(\mathbf{W}_i^t) - \frac{1}{\tilde{K}_i} \mathbf{W}_i^t (\mathbf{W}_i^t)^T . \tag{13}$$

where $\tilde{K}_i = \min\{K, d_i\} \ge \min\{|\mathbb{I}_t \cap \mathcal{N}_i|, d_i\}$. Using similar arguments in Lemma 1, we can show $\mathbf{P}_t$ is positive semi-definite. Therefore,

$$\|\mathbf{z}^{t+1} - \mathbf{z}^t\|_{\mathbf{P}_t}^2 = \sum_{i=1}^{I} \|\mathbf{z}_i^{t+1} - \mathbf{z}_i^t\|_{\mathbf{P}_i^t}^2 = \sum_{i=1}^{I} \|\mathbf{z}_i^{t+1} - \mathbf{z}_i^t\|_{\mathrm{diag}(\mathbf{w}_i^t) - \frac{1}{\tilde{K}_i} \mathbf{w}_i^t (\mathbf{w}_i^t)^T}^2$$

$$= \sum_{i=1}^{I} \left[ \sum_{j_t \in \mathbb{I}_t} \|\mathbf{z}_{ij_t}^{t+1} - \mathbf{z}_{ij_t}^t\|_2^2 - \frac{1}{\tilde{K}_i} \|(\mathbf{w}_i^t)^T (\mathbf{z}_i^{t+1} - \mathbf{z}_i^t)\|_2^2 \right]$$

$$= \sum_{i=1}^{I} \left[ \|\mathbf{z}_i^{t+1} - \mathbf{z}_i^t\|_2^2 - \frac{1}{\tilde{K}_i} \|\mathbf{A}_i^r(\mathbf{x}^{t+1} - \mathbf{x}^t)\|_2^2 \right] . \tag{14}$$

In PDMM, an index set $\mathbb{I}_t$ is randomly chosen. Conditioned on $\mathbf{x}^t$, $\mathbf{x}^{t+1}$ and $\mathbf{y}^{t+1}$ depend on $\mathbb{I}_t$. $\mathbf{P}_t$ depends on $\mathbb{I}_t$. $\mathbf{x}^t, \mathbf{y}^t$ are independent of $\mathbb{I}_t$. $\mathbf{x}^t$ depends on a sequence of observed realization of random variable

$$\xi_{t-1} = \{\mathbb{I}_1, \mathbb{I}_2, \cdots, \mathbb{I}_{t-1}\} . \tag{15}$$

As we do not assume that $f_{j_t}$ is differentiable, we use the subgradient of $f_{j_t}$. In particular, if $f_{j_t}$ is differentiable, the subgradient of $f_{j_t}$ becomes the gradient, i.e., $\nabla f_{j_t}(\mathbf{x}_{j_t})$. PDMM (3)-(5) has the following lemma.

**Lemma 2** *Let $\{\mathbf{x}_{j_t}^t, \mathbf{y}_i^t\}$ be generated by PDMM (3)-(5). Assume $\tau_i > 0$ and $\nu_i \geq 0$. We have*

$$\sum_{j_t \in \mathbb{I}_t} f_{j_t}(\mathbf{x}_{j_t}^{t+1}) - f_{j_t}(\mathbf{x}_{j_t}^*) \leq -\frac{K}{J} \sum_{i=1}^{I} \left\{ \langle \mathbf{y}_i^t, \mathbf{A}_i^r \mathbf{x}^t - \mathbf{a}_i \rangle - \frac{\tau_i \rho}{2} \|\mathbf{A}_i^r \mathbf{x}^t - \mathbf{a}_i\|_2^2 \right\}$$

$$- \sum_{j_t \in \mathbb{I}_t} \langle \hat{\mathbf{y}}^t + \rho(\mathbf{A}\mathbf{x}^t - \mathbf{a}), \mathbf{A}_{j_t}^c(\mathbf{x}_{j_t}^t - \mathbf{x}_{j_t}^*) \rangle + \frac{K}{J} \langle \hat{\mathbf{y}}^t + \rho(\mathbf{A}\mathbf{x}^t - \mathbf{a}), \mathbf{A}\mathbf{x}^t - \mathbf{a} \rangle$$

$$+ \sum_{i=1}^{I} \left\{ \langle \mathbf{y}_i^t, \mathbf{A}_i^r \mathbf{x}^t - \mathbf{a}_i \rangle - \frac{\tau_i \rho}{2} \|\mathbf{A}_i^r \mathbf{x}^t - \mathbf{a}_i\|_2^2 \right\} - \sum_{i=1}^{I} \left\{ \langle \mathbf{y}_i^{t+1}, \mathbf{A}_i^r \mathbf{x}^{t+1} - \mathbf{a}_i \rangle - \frac{\tau_i \rho}{2} \|\mathbf{A}_i^r \mathbf{x}^{t+1} - \mathbf{a}_i\|_2^2 \right\}$$

$$+ \sum_{i=1}^{I} (\|\mathbf{z}^* - \mathbf{z}^t\|_{\mathbf{Q}}^2 - \|\mathbf{z}^* - \mathbf{z}^{t+1}\|_{\mathbf{Q}}^2 - \|\mathbf{z}^{t+1} - \mathbf{z}^t\|_{\mathbf{P}_t}^2)$$

$$+ \sum_{j_t \in \mathbb{I}_t} \eta_{j_t}(B_{\phi_{j_t}}(\mathbf{x}_{j_t}^*, \mathbf{x}_{j_t}^t) - B_{\phi_{j_t}}(\mathbf{x}_{j_t}^*, \mathbf{x}_{j_t}^{t+1}) - B_{\phi_{j_t}}(\mathbf{x}_{j_t}^{t+1}, \mathbf{x}_{j_t}^t))$$

$$+ \frac{\rho}{2} \sum_{i=1}^{I} \left\{ \left[ (1 - \frac{2K}{J})(1 - \nu_i) + (1 - \frac{K}{J})\tau_i + \frac{1}{d_i} \right] \|\mathbf{A}_i^r \mathbf{x}^t - \mathbf{a}_i\|_2^2 - (1 - \nu_i - \tau_i + \frac{1}{d_i}) \|\mathbf{A}_i^r \mathbf{x}^{t+1} - \mathbf{a}_i\|_2^2 \right.$$

$$\left. + (1 - \nu_i - \frac{1}{\tilde{K}_i}) \|\mathbf{A}_i^r(\mathbf{x}^{t+1} - \mathbf{x}^t)\|_2^2 \right\} . \tag{16}$$

*Proof:*    Let $\partial f_{j_t}(\mathbf{x}_{j_t}^{t+1})$ be the subdifferential of $f_{j_t}$ at $\mathbf{x}_{j_t}^{t+1}$. Let $f_j'(\mathbf{x}_j^{t+1}) \in \partial f_j(\mathbf{x}_j^{t+1})$ where $x_j^{t+1} \in \mathcal{X}_j$. For any $\mathbf{x}_{j_t}^* \in \mathcal{X}_{j_t}$, the optimality of the $\mathbf{x}_{j_t}$ update (3) is

$$\langle f_{j_t}'(\mathbf{x}_{j_t}^{t+1}) + (\mathbf{A}_{j_t}^c)^T[\hat{\mathbf{y}}^t + \rho(\mathbf{A}_{j_t}^c \mathbf{x}_{j_t}^{t+1} + \sum_{k \neq j_t} \mathbf{A}_k^c \mathbf{x}_k^t - \mathbf{a})] + \eta_{j_t}(\nabla \phi_{j_t}(\mathbf{x}_{j_t}^{t+1}) - \nabla \phi_{j_t}(\mathbf{x}_{j_t}^t)), \mathbf{x}_{j_t}^{t+1} - \mathbf{x}_{j_t}^* \rangle \leq 0 , \tag{17}$$

Using (5) and rearranging the terms yield

$$\langle f_{j_t}(\mathbf{x}_{j_t}^{t+1}), \mathbf{x}_{j_t}^{t+1} - \mathbf{x}_{j_t}^* \rangle \tag{18}$$
$$\leq \langle -(\mathbf{A}_{j_t}^c)^T[\hat{\mathbf{y}}^t + \rho(\mathbf{A}\mathbf{x}^t - \mathbf{a}) + \rho \mathbf{A}_{j_t}^c(\mathbf{x}_{j_t}^{t+1} - \mathbf{x}_{j_t}^t)] + \eta_{j_t}(\nabla \phi_{j_t}(\mathbf{x}_{j_t}^{t+1}) - \nabla \phi_{j_t}(\mathbf{x}_{j_t}^t)), \mathbf{x}_{j_t}^{t+1} - \mathbf{x}_{j_t}^* \rangle .$$

Using the convexity of $f_{j_t}$, we have

$$
\begin{aligned}
f_{j_t}(\mathbf{x}_{j_t}^{t+1}) - f_{j_t}(\mathbf{x}_{j_t}^*) &\leq -\langle \hat{\mathbf{y}}^t + \rho(\mathbf{A}\mathbf{x}^t - \mathbf{a}), \mathbf{A}_{j_t}^c(\mathbf{x}_{j_t}^{t+1} - \mathbf{x}_{j_t}^*)\rangle \\
&\quad - \rho\langle \mathbf{A}_{j_t}^c(\mathbf{x}_{j_t}^{t+1} - \mathbf{x}_{j_t}^t), \mathbf{A}_{j_t}^c(\mathbf{x}_{j_t}^{t+1} - \mathbf{x}_{j_t}^*)\rangle - \eta_{j_t}\langle \nabla\phi_{j_t}(\mathbf{x}_{j_t}^{t+1}) - \nabla\phi_{j_t}(\mathbf{x}_{j_t}^t), \mathbf{x}_{j_t}^{t+1} - \mathbf{x}_{j_t}^*\rangle \\
&= -\langle \hat{\mathbf{y}}^t + \rho(\mathbf{A}\mathbf{x}^t - \mathbf{a}), \mathbf{A}_{j_t}^c(\mathbf{x}_{j_t}^t - \mathbf{x}_{j_t}^*)\rangle - \langle \hat{\mathbf{y}}^t + \rho(\mathbf{A}\mathbf{x}^t - \mathbf{a}), \mathbf{A}_{j_t}^c(\mathbf{x}_{j_t}^{t+1} - \mathbf{x}_{j_t}^t)\rangle \\
&\quad - \rho\sum_{i=1}^I \langle \mathbf{A}_{ij_t}(\mathbf{x}_{j_t}^{t+1} - \mathbf{x}_{j_t}^t), \mathbf{A}_{ij_t}(\mathbf{x}_{j_t}^{t+1} - \mathbf{x}_{j_t}^*)\rangle \\
&\quad + \eta_{j_t}\left(B_{\phi_{j_t}}(\mathbf{x}_{j_t}^*, \mathbf{x}_{j_t}^t) - B_{\phi_{j_t}}(\mathbf{x}_{j_t}^*, \mathbf{x}_{j_t}^{t+1}) - B_{\phi_{j_t}}(\mathbf{x}_{j_t}^{t+1}, \mathbf{x}_{j_t}^t)\right) .
\end{aligned}
\tag{19}
$$

Summing over $j_t \in \mathbb{I}_t$, we have

$$
\begin{aligned}
&\sum_{j_t \in \mathbb{I}_t} f_{j_t}(\mathbf{x}_{j_t}^{t+1}) - f_{j_t}(\mathbf{x}_{j_t}^*) \\
&\leq -\sum_{j_t \in \mathbb{I}_t}\langle \hat{\mathbf{y}}^t + \rho(\mathbf{A}\mathbf{x}^t - \mathbf{a}), \mathbf{A}_{j_t}^c(\mathbf{x}_{j_t}^t - \mathbf{x}_{j_t}^*)\rangle - \langle \hat{\mathbf{y}}^t + \rho(\mathbf{A}\mathbf{x}^t - \mathbf{a}), \sum_{j_t \in \mathbb{I}_t} \mathbf{A}_{j_t}^c(\mathbf{x}_{j_t}^{t+1} - \mathbf{x}_{j_t}^t)\rangle \\
&\quad - \rho\sum_{i=1}^I \sum_{j_t \in \mathbb{I}_t}\langle \mathbf{A}_{ij_t}(\mathbf{x}_{j_t}^{t+1} - \mathbf{x}_{j_t}^t), \mathbf{A}_{ij_t}(\mathbf{x}_{j_t}^{t+1} - \mathbf{x}_{j_t}^*)\rangle \\
&\quad + \sum_{j_t \in \mathbb{I}_t} \eta_{j_t}\left(B_{\phi_{j_t}}(\mathbf{x}_{j_t}^*, \mathbf{x}_{j_t}^t) - B_{\phi_{j_t}}(\mathbf{x}_{j_t}^*, \mathbf{x}_{j_t}^{t+1}) - B_{\phi_{j_t}}(\mathbf{x}_{j_t}^{t+1}, \mathbf{x}_{j_t}^t)\right) \\
&= -\sum_{j_t \in \mathbb{I}_t}\langle \hat{\mathbf{y}}^t + \rho(\mathbf{A}\mathbf{x}^t - \mathbf{a}), \mathbf{A}_{j_t}^c(\mathbf{x}_{j_t}^t - \mathbf{x}_{j_t}^*)\rangle + \frac{K}{J}\langle \hat{\mathbf{y}}^t + \rho(\mathbf{A}\mathbf{x}^t - \mathbf{a}), \mathbf{A}\mathbf{x}^t - \mathbf{a}\rangle \\
&\quad \underbrace{- \frac{K}{J}\langle \hat{\mathbf{y}}^t + \rho(\mathbf{A}\mathbf{x}^t - \mathbf{a}), \mathbf{A}\mathbf{x}^t - \mathbf{a}\rangle - \langle \hat{\mathbf{y}}^t + \rho(\mathbf{A}\mathbf{x}^t - \mathbf{a}), \mathbf{A}(\mathbf{x}^{t+1} - \mathbf{x}^t)\rangle}_{H_1} \\
&\quad \underbrace{+ \frac{\rho}{2}\sum_{i=1}^I \sum_{j_t \in \mathbb{I}_t}(\|\mathbf{A}_{ij_t}(\mathbf{x}_{j_t}^* - \mathbf{x}_{j_t}^t)\|_2^2 - \|\mathbf{A}_{ij_t}(\mathbf{x}_{j_t}^* - \mathbf{x}_{j_t}^{t+1})\|_2^2 - \|\mathbf{A}_{ij_t}(\mathbf{x}_{j_t}^{t+1} - \mathbf{x}_{j_t}^t)\|_2^2)}_{H_2} \\
&\quad + \sum_{j_t \in \mathbb{I}_t} \eta_{j_t}\left(B_{\phi_{j_t}}(\mathbf{x}_{j_t}^*, \mathbf{x}_{j_t}^t) - B_{\phi_{j_t}}(\mathbf{x}_{j_t}^*, \mathbf{x}_{j_t}^{t+1}) - B_{\phi_{j_t}}(\mathbf{x}_{j_t}^{t+1}, \mathbf{x}_{j_t}^t)\right) .
\end{aligned}
\tag{20}
$$

$H_1$ in (20) can be rewritten as

$$
H_1 = -\langle \hat{\mathbf{y}}^t + \rho(\mathbf{A}\mathbf{x}^t - \mathbf{a}), \mathbf{A}\mathbf{x}^{t+1} - \mathbf{a}\rangle + (1 - \frac{K}{J})\langle \hat{\mathbf{y}}^t + \rho(\mathbf{A}\mathbf{x}^t - \mathbf{a}), \mathbf{A}\mathbf{x}^t - \mathbf{a}\rangle .
\tag{21}
$$

The first term of (21) is equivalent to

$$
\begin{aligned}
&-\langle \hat{\mathbf{y}}^t + \rho(\mathbf{A}\mathbf{x}^t - \mathbf{a}), \mathbf{A}\mathbf{x}^{t+1} - \mathbf{a}\rangle \\
&= -\sum_{i=1}^I \langle \hat{\mathbf{y}}_i^t + \rho(\mathbf{A}_i^r\mathbf{x}^t - \mathbf{a}_i), \mathbf{A}_i^r\mathbf{x}^{t+1} - \mathbf{a}_i\rangle
\end{aligned}
$$

$$= -\sum_{i=1}^{I} \langle \mathbf{y}_i^t + (1 - \nu_i)\rho(\mathbf{A}_i^r \mathbf{x}^t - \mathbf{a}_i), \mathbf{A}_i^r \mathbf{x}^{t+1} - \mathbf{a}_i \rangle$$

$$= -\sum_{i=1}^{I} \left\{ \langle \mathbf{y}_i^{t+1} - \tau_i \rho(\mathbf{A}_i^r \mathbf{x}^{t+1} - \mathbf{a}_i), \mathbf{A}_i^r \mathbf{x}^{t+1} - \mathbf{a}_i \rangle + (1 - \nu_i)\rho\langle \mathbf{A}_i^r \mathbf{x}^t - \mathbf{a}_i, \mathbf{A}_i^r \mathbf{x}^{t+1} - \mathbf{a}_i \rangle \right\}$$

$$= -\sum_{i=1}^{I} \left\{ \langle \mathbf{y}_i^{t+1}, \mathbf{A}_i^r \mathbf{x}^{t+1} - \mathbf{a}_i \rangle - \tau_i \rho \|\mathbf{A}_i^r \mathbf{x}^{t+1} - \mathbf{a}_i\|_2^2 \right.$$
$$\left. - \frac{(1 - \nu_i)\rho}{2}(\|\mathbf{A}_i^r(\mathbf{x}^{t+1} - \mathbf{x}^t)\|_2^2 - \|\mathbf{A}_i^r \mathbf{x}^t - \mathbf{a}_i\|_2^2 - \|\mathbf{A}_i^r \mathbf{x}^{t+1} - \mathbf{a}_i\|_2^2) \right\}$$

$$= -\sum_{i=1}^{I} \left\{ \langle \mathbf{y}_i^{t+1}, \mathbf{A}_i^r \mathbf{x}^{t+1} - \mathbf{a}_i \rangle - \frac{\tau_i \rho}{2} \|\mathbf{A}_i^r \mathbf{x}^{t+1} - \mathbf{a}_i\|_2^2 \right\}$$
$$+ \sum_{i=1}^{I} \left\{ \frac{(1 - \nu_i)\rho}{2}(\|\mathbf{A}_i^r(\mathbf{x}^{t+1} - \mathbf{x}^t)\|_2^2 - \|\mathbf{A}_i^r \mathbf{x}^t - \mathbf{a}_i\|_2^2) - \frac{(1 - \nu_i - \tau_i)\rho}{2} \|\mathbf{A}_i^r \mathbf{x}^{t+1} - \mathbf{a}_i\|_2^2 \right\} . \quad (22)$$

The second term of (21) is equivalent to

$$(1 - \frac{K}{J})\langle \hat{\mathbf{y}}^t + \rho(\mathbf{A}\mathbf{x}^t - \mathbf{a}), \mathbf{A}\mathbf{x}^t - \mathbf{a} \rangle$$

$$= (1 - \frac{K}{J}) \sum_{i=1}^{I} \langle \hat{\mathbf{y}}_i^t + \rho(\mathbf{A}_i^r \mathbf{x}^t - \mathbf{a}_i), \mathbf{A}_i^r \mathbf{x}^t - \mathbf{a}_i \rangle$$

$$= (1 - \frac{K}{J}) \sum_{i=1}^{I} \langle \mathbf{y}_i^t + (1 - \nu_i)\rho(\mathbf{A}_i^r \mathbf{x}^t - \mathbf{a}_i), \mathbf{A}_i^r \mathbf{x}^t - \mathbf{a}_i \rangle$$

$$= (1 - \frac{K}{J}) \sum_{i=1}^{I} \left\{ \langle \mathbf{y}_i^t, \mathbf{A}_i^r \mathbf{x}^t - \mathbf{a}_i \rangle - \frac{\tau_i \rho}{2} \|\mathbf{A}_i^r \mathbf{x}^t - \mathbf{a}_i\|_2^2 \right\} + (1 - \frac{K}{J}) \sum_{i=1}^{I} (1 - \nu_i + \frac{\tau_i}{2})\rho\|\mathbf{A}_i^r \mathbf{x}^t - \mathbf{a}_i\|_2^2 .$$
$$(23)$$

$H_2$ in (20) is equavilant to

$$H_2 = \frac{\rho}{2} \sum_{i=1}^{I} \sum_{j_t \in \mathbb{I}_t} (\|\mathbf{z}_{ij_t}^* - \mathbf{z}_{ij_t}^t\|_2^2 - \|\mathbf{z}_{ij_t}^* - \mathbf{z}_{ij_t}^{t+1}\|_2^2 - \|\mathbf{z}_{ij_t}^{t+1} - \mathbf{z}_{ij_t}^t\|_2^2)$$

$$= \frac{\rho}{2} \sum_{i=1}^{I} (\|\mathbf{z}_i^* - \mathbf{z}_i^t\|_2^2 - \|\mathbf{z}_i^* - \mathbf{z}_i^{t+1}\|_2^2 - \|\mathbf{z}_i^{t+1} - \mathbf{z}_i^t\|_2^2)$$

$$= \frac{\rho}{2} (\|\mathbf{z}^* - \mathbf{z}^t\|_{\mathbf{Q}}^2 - \|\mathbf{z}^* - \mathbf{z}^{t+1}\|_{\mathbf{Q}}^2 - \|\mathbf{z}^{t+1} - \mathbf{z}^t\|_{\mathbf{P}_t}^2)$$

$$+ \frac{\rho}{2} \sum_{i=1}^{I} \frac{1}{d_i}(\|\mathbf{A}_i^r \mathbf{x}^t - \mathbf{a}_i\|_2^2 - \|\mathbf{A}_i^r \mathbf{x}^{t+1} - \mathbf{a}_i\|_2^2) - \frac{1}{\tilde{K}_i} \|\mathbf{A}_i^r(\mathbf{x}^{t+1} - \mathbf{x}^t)\|_2^2 . \quad (24)$$

where the last equality uses the definition of $\mathbf{Q}$ in (7) and $\mathbf{P}_t$ (13), and $\tilde{K}_i = \min\{K, d_i\}$. Combining the results of (21)-(24) gives

$$H_1 + H_2 = -\sum_{i=1}^{I} \left\{ \langle \mathbf{y}_i^{t+1}, \mathbf{A}_i^r \mathbf{x}^{t+1} - \mathbf{a}_i \rangle - \frac{\tau_i \rho}{2} \|\mathbf{A}_i^r \mathbf{x}^{t+1} - \mathbf{a}_i\|_2^2 \right\}$$

$$+\sum_{i=1}^{I}\left\{\frac{(1-\nu_i)\rho}{2}(\|\mathbf{A}_i^r(\mathbf{x}^{t+1}-\mathbf{x}^t)\|_2^2-\|\mathbf{A}_i^r\mathbf{x}^t-\mathbf{a}_i\|_2^2)-\frac{(1-\nu_i-\tau_i)\rho}{2}\|\mathbf{A}_i^r\mathbf{x}^{t+1}-\mathbf{a}_i\|_2^2\right\}$$

$$+(1-\frac{K}{J})\sum_{i=1}^{I}\left\{\langle\mathbf{y}_i^t,\mathbf{A}_i^r\mathbf{x}^t-\mathbf{a}_i\rangle-\frac{\tau_i\rho}{2}\|\mathbf{A}_i^r\mathbf{x}^t-\mathbf{a}_i\|_2^2\right\}+(1-\frac{K}{J})\sum_{i=1}^{I}(1-\nu_i+\frac{\tau_i}{2})\rho\|\mathbf{A}_i^r\mathbf{x}^t-\mathbf{a}_i\|_2^2$$

$$+\frac{\rho}{2}(\|\mathbf{z}^*-\mathbf{z}^t\|_{\mathbf{Q}}^2-\|\mathbf{z}^*-\mathbf{z}^{t+1}\|_{\mathbf{Q}}^2-\|\mathbf{z}^{t+1}-\mathbf{z}^t\|_{\mathbf{P}_t}^2)$$

$$+\frac{\rho}{2}\sum_{i=1}^{I}\frac{1}{d_i}(\|\mathbf{A}_i^r\mathbf{x}^t-\mathbf{a}_i\|_2^2-\|\mathbf{A}_i^r\mathbf{x}^{t+1}-\mathbf{a}_i\|_2^2)-\frac{1}{\tilde{K}_i}\|\mathbf{A}_i^r(\mathbf{x}^{t+1}-\mathbf{x}^t)\|_2^2)$$

$$=-\frac{K}{J}\sum_{i=1}^{I}\left\{\langle\mathbf{y}_i^t,\mathbf{A}_i^r\mathbf{x}^t-\mathbf{a}_i\rangle-\frac{\tau_i\rho}{2}\|\mathbf{A}_i^r\mathbf{x}^t-\mathbf{a}_i\|_2^2\right\}$$

$$+\sum_{i=1}^{I}\left\{\langle\mathbf{y}_i^t,\mathbf{A}_i^r\mathbf{x}^t-\mathbf{a}_i\rangle-\frac{\tau_i\rho}{2}\|\mathbf{A}_i^r\mathbf{x}^t-\mathbf{a}_i\|_2^2\right\}-\sum_{i=1}^{I}\left\{\langle\mathbf{y}_i^{t+1},\mathbf{A}_i^r\mathbf{x}^{t+1}-\mathbf{a}_i\rangle-\frac{\tau_i\rho}{2}\|\mathbf{A}_i^r\mathbf{x}^{t+1}-\mathbf{a}_i\|_2^2\right\}$$

$$+\frac{\rho}{2}(\|\mathbf{z}^*-\mathbf{z}^t\|_{\mathbf{Q}}^2-\|\mathbf{z}^*-\mathbf{z}^{t+1}\|_{\mathbf{Q}}^2-\|\mathbf{z}^{t+1}-\mathbf{z}^t\|_{\mathbf{P}_t}^2)$$

$$+\frac{\rho}{2}\sum_{i=1}^{I}\left\{[(1-\frac{2K}{J})(1-\nu_i)+(1-\frac{K}{J})\tau_i+\frac{1}{d_i}]\|\mathbf{A}_i^r\mathbf{x}^t-\mathbf{a}_i\|_2^2-(1-\nu_i-\tau_i+\frac{1}{d_i})\|\mathbf{A}_i^r\mathbf{x}^{t+1}-\mathbf{a}_i\|_2^2\right.$$

$$\left.++(1-\nu_i-\frac{1}{\tilde{K}_i})\|\mathbf{A}_i^r(\mathbf{x}^{t+1}-\mathbf{x}^t)\|_2^2\right\}\ . \tag{25}$$

Plugging back into (20) completes the proof. ∎

**Lemma 3** *Let $\{\mathbf{x}_{j_t}^t,\mathbf{y}_i^t\}$ be generated by PDMM (3)-(5). Assume $\tau_i>0$ and $\nu_i\geq 0$. We have*

$$\sum_{j_t\in\mathbb{I}_t}f_{j_t}(\mathbf{x}_{j_t}^{t+1})-f_{j_t}(\mathbf{x}_{j_t}^*)\leq-\frac{K}{J}\sum_{i=1}^{I}\left\{\langle\mathbf{y}_i^t,\mathbf{A}_i^r\mathbf{x}^t-\mathbf{a}_i\rangle-\frac{\tau_i\rho}{2}\|\mathbf{A}_i^r\mathbf{x}^t-\mathbf{a}_i\|_2^2\right\}$$

$$-\sum_{j_t\in\mathbb{I}_t}\langle\hat{\mathbf{y}}^t+\rho(\mathbf{A}\mathbf{x}^t-\mathbf{a}),\mathbf{A}_{j_t}^c(\mathbf{x}_{j_t}^t-\mathbf{x}_{j_t}^*)\rangle+\frac{K}{J}\langle\hat{\mathbf{y}}^t+\rho(\mathbf{A}\mathbf{x}^t-\mathbf{a}),\mathbf{A}\mathbf{x}^t-\mathbf{a}\rangle$$

$$+\sum_{i=1}^{I}\left\{\langle\mathbf{y}_i^t,\mathbf{A}_i^r\mathbf{x}^t-\mathbf{a}_i\rangle-\frac{\tau_i\rho}{2}\|\mathbf{A}_i^r\mathbf{x}^t-\mathbf{a}_i\|_2^2\right\}-\sum_{i=1}^{I}\left\{\langle\mathbf{y}_i^{t+1},\mathbf{A}_i^r\mathbf{x}^{t+1}-\mathbf{a}_i\rangle-\frac{\tau_i\rho}{2}\|\mathbf{A}_i^r\mathbf{x}^{t+1}-\mathbf{a}_i\|_2^2\right\}$$

$$+\|\mathbf{z}^*-\mathbf{z}^t\|_{\mathbf{Q}}^2-\|\mathbf{z}^*-\mathbf{z}^{t+1}\|_{\mathbf{Q}}^2-\|\mathbf{z}^{t+1}-\mathbf{z}^t\|_{\mathbf{P}_t}^2$$

$$+\boldsymbol{\eta}^T(B_\phi(\mathbf{x}^*,\mathbf{x}^t)-B_\phi(\mathbf{x}^*,\mathbf{x}^{t+1})-B_\phi(\mathbf{x}^{t+1},\mathbf{x}^t))$$

$$+\frac{\rho}{2}\sum_{i=1}^{I}\left[\gamma_i(\|\mathbf{A}_i^r\mathbf{x}^t-\mathbf{a}_i\|_2^2-\|\mathbf{A}_i^r\mathbf{x}^{t+1}-\mathbf{a}_i\|_2^2)-\beta_i\|\mathbf{A}_i^r\mathbf{x}^{t+1}-\mathbf{a}_i\|_2^2\right]\ . \tag{26}$$

*where $\boldsymbol{\eta}^T=[\eta_1,\cdots,\eta_J]$. $\tau_i>0,\nu_i\geq 0,\gamma_i\geq 0$ and $\beta_i\geq 0$ satisfy the following conditions:*

$$\nu_i\in(\max\{0,1-\frac{2J}{\tilde{K}_i(2J-K)}\},1-\frac{1}{\tilde{K}_i}]\ , \tag{27}$$

$$\tau_i \leq \frac{J}{2J-K}[\frac{4}{\tilde{K}_i} - (4 - \frac{2K}{J})(1 - \nu_i)] \leq \frac{2K}{\tilde{K}_i(2J-K)} \ , \tag{28}$$

$$\gamma_i = (3 - \frac{2K}{J})(1 - \nu_i) + (1 - \frac{K}{J})\tau_i + \frac{1}{d_i} - \frac{2}{\tilde{K}_i} \ , \tag{29}$$

$$\beta_i = \frac{4}{\tilde{K}_i} - (2 - \frac{K}{J})[2(1 - \nu_i) + \tau_i] \ . \tag{30}$$

*Proof:* In (16), denote

$$H_3 = [(1 - \frac{2K}{J})(1 - \nu_i) + (1 - \frac{K}{J})\tau_i + \frac{1}{d_i}]\|\mathbf{A}_i^r \mathbf{x}^t - \mathbf{a}_i\|_2^2 - (1 - \nu_i - \tau_i + \frac{1}{d_i})\|\mathbf{A}_i^r \mathbf{x}^{t+1} - \mathbf{a}_i\|_2^2 \ , \tag{31}$$

$$H_4 = (1 - \nu_i - \frac{1}{\tilde{K}_i})\|\mathbf{A}_i^r(\mathbf{x}^{t+1} - \mathbf{x}^t)\|_2^2 \ . \tag{32}$$

Our goal is to eliminate $H_4$ so that

$$H_3 + H_4 = \gamma_i(\|\mathbf{A}_i^r \mathbf{x}^t - \mathbf{a}_i\|_2^2 - \|\mathbf{A}_i^r \mathbf{x}^{t+1} - \mathbf{a}_i\|_2^2) - \beta_i\|\mathbf{A}_i^r \mathbf{x}^{t+1} - \mathbf{a}_i\|_2^2 \ , \tag{33}$$

where $\gamma_i \geq 0$ and $\beta_i \geq 0$ .

We want to choose a large $\tau_i$ and a small $\nu_i$. Assume $1 - \nu_i - \frac{1}{\tilde{K}_i} \geq 0$, i.e., $\nu_i \leq 1 - \frac{1}{\tilde{K}_i}$, we have

$$H_4 = (1 - \nu_i - \frac{1}{\tilde{K}_i})\|\mathbf{A}_i^r(\mathbf{x}^{t+1} - \mathbf{x}^t)\|_2^2 \leq 2(1 - \nu_i - \frac{1}{\tilde{K}_i})(\|\mathbf{A}_i^r \mathbf{x}^t - \mathbf{a}_i\|_2^2 + \|\mathbf{A}_i^r \mathbf{x}^{t+1} - \mathbf{a}_i\|_2^2) \ . \tag{34}$$

Therefore, we have

$$H_3 + H_4 \leq [(3 - \frac{2K}{J})(1 - \nu_i) + (1 - \frac{K}{J})\tau_i + \frac{1}{d_i} - \frac{2}{\tilde{K}_i}]\|\mathbf{A}_i^r \mathbf{x}^t - \mathbf{a}_i\|_2^2 + (1 - \nu_i + \tau_i - \frac{1}{d_i} - \frac{2}{\tilde{K}_i})\|\mathbf{A}_i^r \mathbf{x}^{t+1} - \mathbf{a}_i\|_2^2$$
$$= \gamma_i(\|\mathbf{A}_i^r \mathbf{x}^t - \mathbf{a}_i\|_2^2 - \|\mathbf{A}_i^r \mathbf{x}^{t+1} - \mathbf{a}_i\|_2^2) - \beta_i\|\mathbf{A}_i^r \mathbf{x}^{t+1} - \mathbf{a}_i\|_2^2 \ . \tag{35}$$

where

$$\gamma_i = (3 - \frac{2K}{J})(1 - \nu_i) + (1 - \frac{K}{J})\tau_i + \frac{1}{d_i} - \frac{2}{\tilde{K}_i}$$
$$\geq (3 - \frac{2K}{J})\frac{1}{\tilde{K}_i} + (1 - \frac{K}{J})\tau_i + \frac{1}{d_i} - \frac{2}{\tilde{K}_i}$$
$$= (1 - \frac{K}{J})\frac{1}{\tilde{K}_i} - \frac{K}{J\tilde{K}_i} + \frac{1}{d_i} + (1 - \frac{K}{J})\tau_i \geq 0 \ . \tag{36}$$

and

$$\beta_i = -(1 - \nu_i + \tau_i + \frac{1}{d_i} - \frac{2}{\tilde{K}_i} + \gamma_i) = \frac{4}{\tilde{K}_i} - (2 - \frac{K}{J})[2(1 - \nu_i) + \tau_i] \ . \tag{37}$$

We also want $\beta_i \geq 0$, which can be reduced to

$$\tau_i \leq \frac{J}{2J-K}[\frac{4}{\tilde{K}_i} - (4 - \frac{2K}{J})(1 - \nu_i)] \tag{38}$$

$$\leq \frac{J}{2J-K}[\frac{4}{\tilde{K}_i} - (4 - \frac{2K}{J})\frac{1}{\tilde{K}_i}]$$

$$= \frac{2K}{\tilde{K}_i(2J-K)} .$$

It also requires the RHS of (38) to be positive, leading to $\nu_i > \max\{0, 1 - \frac{2J}{\tilde{K}_i(2J-K)}\}$. Therefore, $\nu_i \in (\max\{0, 1 - \frac{2J}{\tilde{K}_i(2J-K)}\}, 1 - \frac{1}{\tilde{K}_i}]$.

Denote $B_\phi = [B_{\phi_1}, \cdots, B_{\phi_J}]^T$ as a column vector of the Bregman divergence on block coordinates of $\mathbf{x}$. Using $\mathbf{x}^{t+1} = [\mathbf{x}^{t+1}_{j_t \in \mathbb{I}_t}, \mathbf{x}^t_{j_t \notin \mathbb{I}_t}]^T$, we have $B_{\phi_{j_t}}(\mathbf{x}^*_{j_t}, \mathbf{x}^t_{j_t}) - B_{\phi_{j_t}}(\mathbf{x}^*_{j_t}, \mathbf{x}^{t+1}_{j_t}) = B_\phi(\mathbf{x}^*, \mathbf{x}^t) - B_\phi(\mathbf{x}^*, \mathbf{x}^{t+1}), B_{\phi_{j_t}}(\mathbf{x}^{t+1}_{j_t}, \mathbf{x}^t_{j_t}) = B_\phi(\mathbf{x}^{t+1}, \mathbf{x}^t)$. Thus,

$$\sum_{j_t \in \mathbb{I}_t} \eta_{j_t} \left( B_{\phi_{j_t}}(\mathbf{x}^*_{j_t}, \mathbf{x}^t_{j_t}) - B_{\phi_{j_t}}(\mathbf{x}^*_{j_t}, \mathbf{x}^{t+1}_{j_t}) - B_{\phi_{j_t}}(\mathbf{x}^{t+1}_{j_t}, \mathbf{x}^t_{j_t}) \right)$$
$$= \boldsymbol{\eta}^T(B_\phi(\mathbf{x}^*, \mathbf{x}^t) - B_\phi(\mathbf{x}^*, \mathbf{x}^{t+1}) - B_\phi(\mathbf{x}^{t+1}, \mathbf{x}^t)) . \tag{39}$$

where $\boldsymbol{\eta}^T = [\eta_1, \cdots, \eta_J]$. ∎

**Lemma 4** *Let $\{\mathbf{x}^t_{j_t}, \mathbf{y}^t_i\}$ be generated by PDMM (3)-(5). Assume $\tau_i > 0$ and $\nu_i \geq 0$ satisfy the conditions in Lemma 3. We have*

$$f(\mathbf{x}^t) - f(\mathbf{x}^*) \leq -\sum_{i=1}^I \left\{ \langle \mathbf{y}^t_i, \mathbf{A}^r_i \mathbf{x}^t - \mathbf{a}_i \rangle - \frac{\tau_i \rho}{2} \|\mathbf{A}^r_i \mathbf{x}^t - \mathbf{a}_i\|_2^2 \right\}$$

$$+ \frac{J}{K} \left\{ \tilde{\mathcal{L}}_\rho(\mathbf{x}^t, \mathbf{y}^t) - \mathbb{E}_{\mathbb{I}_t} \tilde{\mathcal{L}}_\rho(\mathbf{x}^{t+1}, \mathbf{y}^{t+1}) - \frac{\rho}{2} \sum_{i=1}^I \beta_i \mathbb{E}_{\mathbb{I}_t} \|\mathbf{A}^r_i \mathbf{x}^{t+1} - \mathbf{a}_i\|_2^2 \right.$$

$$+ \frac{\rho}{2}(\|\mathbf{z}^* - \mathbf{z}^t\|^2_{\mathbf{Q}} - \mathbb{E}_{\mathbb{I}_t}\|\mathbf{z}^* - \mathbf{z}^{t+1}\|^2_{\mathbf{Q}} - \mathbb{E}_{\mathbb{I}_t}\|\mathbf{z}^{t+1} - \mathbf{z}^t\|^2_{\mathbf{P}_t})$$

$$\left. + \boldsymbol{\eta}^T(B_\phi(\mathbf{x}^*, \mathbf{x}^t) - \mathbb{E}_{\mathbb{I}_t} B_\phi(\mathbf{x}^*, \mathbf{x}^{t+1}) - \mathbb{E}_{\mathbb{I}_t} B_\phi(\mathbf{x}^{t+1}, \mathbf{x}^t)) \right\} . \tag{40}$$

*where $\tilde{\mathcal{L}}_\rho$ is defined as follows:*

$$\tilde{\mathcal{L}}_\rho(\mathbf{x}^t, \mathbf{y}^t) = f(\mathbf{x}^t) - f(\mathbf{x}^*) + \sum_{i=1}^I \left\{ \langle \mathbf{y}^t_i, \mathbf{A}^r_i \mathbf{x}^t - \mathbf{a}_i \rangle + \frac{(\gamma_i - \tau_i)\rho}{2} \|\mathbf{A}^r_i \mathbf{x}^t - \mathbf{a}_i\|_2^2 \right\} . \tag{41}$$

*$\tau_i, \nu_i, \gamma_i, \beta_i$ and $\boldsymbol{\eta}$ are defined in Lemma 3.*

*Proof:* Using $\mathbf{x}^{t+1} = [\mathbf{x}^{t+1}_{j_t \in \mathbb{I}_t}, \mathbf{x}^t_{j_t \notin \mathbb{I}_t}]^T$, we have

$$f(\mathbf{x}^{t+1}) - f(\mathbf{x}^t) = \sum_{j_t \in \mathbb{I}_t} f_{j_t}(\mathbf{x}^{t+1}_{j_t}) - f_{j_t}(\mathbf{x}^t_{j_t}) = \sum_{j_t \in \mathbb{I}_t}[f_{j_t}(\mathbf{x}^{t+1}_{j_t}) - f_{j_t}(\mathbf{x}^*_{j_t})] - \sum_{j_t \in \mathbb{I}_t}[f_{j_t}(\mathbf{x}^t_{j_t}) - f_{j_t}(\mathbf{x}^*_{j_t})] . \tag{42}$$

Rearranging the terms and using Lemma 3 yield

$$\sum_{j_t \in \mathbb{I}_t} f_{j_t}(\mathbf{x}^t_{j_t}) - f_{j_t}(\mathbf{x}^*_{j_t}) = \sum_{j \in \mathbb{I}_t}[f_{j_t}(\mathbf{x}^{t+1}_{j_t}) - f_{j_t}(\mathbf{x}^*_{j_t})] + f(\mathbf{x}^t) - f(\mathbf{x}^{t+1})$$

$$\leq -\frac{K}{J}\sum_{i=1}^{I}\left\{\langle \mathbf{y}_i^t, \mathbf{A}_i^r\mathbf{x}^t - \mathbf{a}_i\rangle - \frac{\tau_i\rho}{2}\|\mathbf{A}_i^r\mathbf{x}^t - \mathbf{a}_i\|_2^2\right\}$$

$$- \sum_{j_t\in\mathbb{I}_t}\langle \hat{\mathbf{y}}^t + \rho(\mathbf{A}\mathbf{x}^t - \mathbf{a}), \mathbf{A}_{j_t}^c(\mathbf{x}_{j_t}^t - \mathbf{x}_{j_t}^*)\rangle + \frac{K}{J}\langle \hat{\mathbf{y}}^t + \rho(\mathbf{A}\mathbf{x}^t - \mathbf{a}), \mathbf{A}\mathbf{x}^t - \mathbf{a}\rangle$$

$$+ \tilde{\mathcal{L}}_\rho(\mathbf{x}^t, \mathbf{y}^t) - \tilde{\mathcal{L}}_\rho(\mathbf{x}^{t+1}, \mathbf{y}^{t+1}) - \frac{\rho}{2}\sum_{i=1}^{I}\beta_i\|\mathbf{A}_i^r\mathbf{x}^{t+1} - \mathbf{a}_i\|_2^2$$

$$+ \sum_{i=1}^{I}(\|\mathbf{z}^* - \mathbf{z}^t\|_{\mathbf{Q}}^2 - \|\mathbf{z}^* - \mathbf{z}^{t+1}\|_{\mathbf{Q}}^2 - \|\mathbf{z}^{t+1} - \mathbf{z}^t\|_{\mathbf{P}_t}^2)$$

$$+ \sum_{j_t\in\mathbb{I}_t}\left[\boldsymbol{\eta}^T(B_\phi(\mathbf{x}^*, \mathbf{x}^t) - B_\phi(\mathbf{x}^*, \mathbf{x}^{t+1}) - B_\phi(\mathbf{x}^{t+1}, \mathbf{x}^t))\right] , \tag{43}$$

where $\tilde{\mathcal{L}}_\rho(\mathbf{x}^t, \mathbf{y}^t)$ is defined in (41). Conditioning on $\mathbf{x}^t$ and taking expectation over $\mathbb{I}_t$, we have

$$\frac{K}{J}[f(\mathbf{x}^t) - f(\mathbf{x}^*)] \leq -\frac{K}{J}\sum_{i=1}^{I}\left\{\langle \mathbf{y}_i^t, \mathbf{A}_i^r\mathbf{x}^t - \mathbf{a}_i\rangle - \frac{\tau_i\rho}{2}\|\mathbf{A}_i^r\mathbf{x}^t - \mathbf{a}_i\|_2^2\right\}$$

$$+ \tilde{\mathcal{L}}_\rho(\mathbf{x}^t, \mathbf{y}^t) - \mathbb{E}_{\mathbb{I}_t}\tilde{\mathcal{L}}_\rho(\mathbf{x}^{t+1}, \mathbf{y}^{t+1}) - \frac{\rho}{2}\sum_{i=1}^{I}\beta_i\mathbb{E}_{\mathbb{I}_t}\|\mathbf{A}_i^r\mathbf{x}^{t+1} - \mathbf{a}_i\|_2^2$$

$$+ \frac{\rho}{2}(\|\mathbf{z}^* - \mathbf{z}^t\|_{\mathbf{Q}}^2 - \mathbb{E}_{\mathbb{I}_t}\|\mathbf{z}^* - \mathbf{z}^{t+1}\|_{\mathbf{Q}}^2 - \mathbb{E}_{\mathbb{I}_t}\|\mathbf{z}^{t+1} - \mathbf{z}^t\|_{\mathbf{P}_t}^2)$$

$$+ \sum_{j_t\in\mathbb{I}_t}\left[\boldsymbol{\eta}^T(B_\phi(\mathbf{x}^*, \mathbf{x}^t) - \mathbb{E}_{\mathbb{I}_t}B_\phi(\mathbf{x}^*, \mathbf{x}^{t+1}) - \mathbb{E}_{\mathbb{I}_t}B_\phi(\mathbf{x}^{t+1}, \mathbf{x}^t))\right] , \tag{44}$$

where we use

$$\mathbb{E}_{\mathbb{I}_t}[-\sum_{j_t\in\mathbb{I}_t}\langle \hat{\mathbf{y}}^t + \rho(\mathbf{A}\mathbf{x}^t - \mathbf{a}), \mathbf{A}_{j_t}^c(\mathbf{x}_{j_t}^t - \mathbf{x}_{j_t}^*)\rangle] = -\frac{K}{J}\langle \hat{\mathbf{y}}^t + \rho(\mathbf{A}\mathbf{x}^t - \mathbf{a}), \mathbf{A}\mathbf{x}^t - \mathbf{a}\rangle . \tag{45}$$

Dividing both sides by $\frac{K}{J}$ and using the definition (41) complete the proof. ∎

## 1.2 Theoretical Results

We establish the convergence results for PDMM under fairly simple assumptions:

**Assumption 1**
 *(1) $f_j : \mathbb{R}^{n_j} \to \mathbb{R} \cup \{+\infty\}$ are closed, proper, and convex.*
 *(2) A KKT point of the Lagrangian ($\rho = 0$ in (2)) of Problem (1) exists.*

Assumption 1 is the same as that required by ADMM [1, 4]. Let $\partial f_j$ be the subdifferential of $f_j$. Assume that $\{\mathbf{x}_j^* \in \mathcal{X}_j, \mathbf{y}_i^*\}$ satisfies the KKT conditions of the Lagrangian ($\rho = 0$ in (2)), i.e.,

$$-\mathbf{A}_j^T\mathbf{y}^* \in \partial f_j(\mathbf{x}_j^*) , \tag{46}$$

$$\mathbf{A}\mathbf{x}^* - \mathbf{a} = 0. \tag{47}$$

During iterations, (47) is satisfied if $\mathbf{A}\mathbf{x}^{t+1} = \mathbf{a}$. Let $f_j'(\mathbf{x}_j^{t+1}) \in \partial f_j(\mathbf{x}_j^{t+1})$ where $x_j^{t+1} \in \mathcal{X}_j$. For any $\mathbf{x}_j \in \mathcal{X}_j$, the optimality conditions for the $\mathbf{x}_j$ update (3) is

$$\langle f_j'(\mathbf{x}_j^{t+1}) + \mathbf{A}_j^c[\hat{\mathbf{y}}^t + \rho(\mathbf{A}_j^c\mathbf{x}_j^{t+1} + \sum_{k \neq j} \mathbf{A}_k^c\mathbf{x}_k^t - \mathbf{a})] + \eta_j(\nabla\phi_j(\mathbf{x}_j^{t+1}) - \nabla\phi_j(\mathbf{x}_j^t)), \mathbf{x}_j^{t+1} - \mathbf{x}_j \rangle \leq 0 \,, \tag{48}$$

which is sufficiently satisfied if

$$-\mathbf{A}_j^c[\mathbf{y}^t + (1 - \nu)\rho(\mathbf{A}\mathbf{x}^t - \mathbf{a}) + \mathbf{A}_j^c(\mathbf{x}_j^{t+1} - \mathbf{x}_j^t)] - \eta_j(\nabla\phi_j(\mathbf{x}_j^{t+1}) - \nabla\phi_j(\mathbf{x}_j^t)) = f_j'(\mathbf{x}_j^{t+1}) \,. \tag{49}$$

When $\mathbf{A}\mathbf{x}^{t+1} = \mathbf{a}$, $\mathbf{y}^{t+1} = \mathbf{y}^t$. If $\mathbf{A}_j^c(\mathbf{x}_j^{t+1} - \mathbf{x}_j^t) = 0$, then $\mathbf{A}\mathbf{x}^t - \mathbf{a} = 0$. When $\eta_j \geq 0$, further assuming $B_{\phi_j}(\mathbf{x}_j^{t+1}, \mathbf{x}_j^t) = 0$, (46) will be satisfied. Overall, the KKT conditions (46)-(47) are satisfied if the following optimality conditions are satisfied by the iterates:

$$\mathbf{A}\mathbf{x}^{t+1} = \mathbf{a} \,, \mathbf{A}_j^c(\mathbf{x}_j^{t+1} - \mathbf{x}_j^t) = 0 \,, \tag{50}$$

$$B_{\phi_j}(\mathbf{x}_j^{t+1}, \mathbf{x}_j^t) = 0 \,. \tag{51}$$

The above optimality conditions are sufficient for the KKT conditions. (50) are the optimality conditions for the exact PDMM. (51) is needed only when $\eta_j > 0$.

In Lemma 3, setting the values of $\nu_i, \tau_i, \gamma_i, \beta_i$ as follows:

$$\nu_i = 1 - \frac{1}{\tilde{K}_i} \,, \tau_i = \frac{K}{\tilde{K}_i(2J - K)} \,, \gamma_i = \frac{2(J - K)}{\tilde{K}_i(2J - K)} + \frac{1}{d_i} - \frac{K}{J\tilde{K}_i} \,, \beta_i = \frac{K}{J\tilde{K}_i} \,. \tag{52}$$

Define the residual of optimality conditions (50)-(51) as

$$R(\mathbf{x}^{t+1}) = \frac{\rho}{2}\|\mathbf{z}^{t+1} - \mathbf{z}^t\|_{\mathbf{P}_t}^2 + \frac{\rho}{2}\sum_{i=1}^I \beta_i\|\mathbf{A}_i^r\mathbf{x}^{t+1} - \mathbf{a}_i\|_2^2 + [\boldsymbol{\eta}^T B_\phi(\mathbf{x}^{t+1}, \mathbf{x}^t)] \,. \tag{53}$$

If $R(\mathbf{x}^{t+1}) \to 0$, (50)-(51) will be satisfied and thus PDMM converges to the KKT point $\{\mathbf{x}^*, \mathbf{y}^*\}$.

Define the current iterate $\mathbf{v}^t = (\mathbf{x}_j^t, \mathbf{y}_i^t)$ and $h(\mathbf{v}^*, \mathbf{v}^t)$ as a distance from $\mathbf{v}^t$ to a KKT point $\mathbf{v}^* = (\mathbf{x}_j^*, \mathbf{y}_i^*)$:

$$h(\mathbf{v}^*, \mathbf{v}^t) = \frac{K}{J}\sum_{i=1}^I \frac{1}{2\tau_i\rho}\|\mathbf{y}_i^* - \mathbf{y}_i^{t-1}\|_2^2 + \tilde{\mathcal{L}}_\rho(\mathbf{x}^t, \mathbf{y}^t) + \frac{\rho}{2}\|\mathbf{z}^* - \mathbf{z}^t\|_{\mathbf{Q}}^2 + \boldsymbol{\eta}^T B_\phi(\mathbf{x}^*, \mathbf{x}^t) \,. \tag{54}$$

The following Lemma shows that $h(\mathbf{v}^*, \mathbf{v}^t) \geq 0$.

**Lemma 5** *Let $h(\mathbf{v}^*, \mathbf{v}^t)$ be defined in (54). Setting $\nu_i = 1 - \frac{1}{\tilde{K}_i}$ and $\tau_i = \frac{K}{\tilde{K}_i(2J-K)}$, we have*

$$h(\mathbf{v}^*, \mathbf{v}^t) \geq \frac{\rho}{2}\sum_{i=1}^I \zeta_i\|\mathbf{A}_i^r\mathbf{x}^t - \mathbf{a}_i\|_2^2 + \frac{\rho}{2}\|\mathbf{z}^* - \mathbf{z}^t\|_{\mathbf{Q}}^2 + +\sum_{j=1}^J \eta_j B_{\phi_j}(\mathbf{x}_j^*, \mathbf{x}_j^t) \geq 0 \,. \tag{55}$$

*where $\zeta_i = \frac{J-K}{\tilde{K}_i(2J-K)} + \frac{1}{d_i} - \frac{K}{J\tilde{K}_i} \geq 0$. Moreover, if $h(\mathbf{v}^*, \mathbf{v}^t) = 0$, then $\mathbf{A}_i^r\mathbf{x}^t = \mathbf{a}_i, \mathbf{z}^t = \mathbf{z}^*$ and $B_{\phi_j}(\mathbf{x}_j^*, \mathbf{x}_j^t) = 0$. Thus, (46)-(47) are satisfied.*

*Proof:*   Using the convexity of $f$ and (46), we have

$$f(\mathbf{x}^*) - f(\mathbf{x}^t) \leq -\langle \mathbf{A}^T \mathbf{y}^*, \mathbf{x}^* - \mathbf{x}^t \rangle = \sum_{i=1}^{I} \langle \mathbf{y}_i^*, \mathbf{A}_i^r \mathbf{x}^t - \mathbf{a}_i \rangle . \tag{56}$$

Thus,

$$\tilde{\mathcal{L}}_\rho(\mathbf{x}^t, \mathbf{y}^t) = f(\mathbf{x}^t) - f(\mathbf{x}^*) + \sum_{i=1}^{I} \left\{ \langle \mathbf{y}_i^t, \mathbf{A}_i^r \mathbf{x}^t - \mathbf{a}_i \rangle + \frac{(\gamma_i - \tau_i)\rho}{2} \|\mathbf{A}_i^r \mathbf{x}^t - \mathbf{a}_i\|_2^2 \right\}$$

$$\geq \sum_{i=1}^{I} \left\{ \langle \mathbf{y}_i^t - \mathbf{y}_i^*, \mathbf{A}_i^r \mathbf{x}^t - \mathbf{a}_i \rangle + \frac{(\gamma_i - \tau_i)\rho}{2} \|\mathbf{A}_i^r \mathbf{x}^t - \mathbf{a}_i\|_2^2 \right\}$$

$$= \sum_{i=1}^{I} \left\{ \langle \mathbf{y}_i^{t-1} - \mathbf{y}_i^*, \mathbf{A}_i \mathbf{x}^t - \mathbf{a}_i \rangle + \langle \mathbf{y}_i^t - \mathbf{y}_i^{t-1}, \mathbf{A}_i \mathbf{x}^t - \mathbf{a}_i \rangle + \frac{(\gamma_i - \tau_i)\rho}{2} \|\mathbf{A}_i^r \mathbf{x}^t - \mathbf{a}_i\|_2^2 \right\}$$

$$\geq \sum_{i=1}^{I} \left[ -\frac{K}{2J\tau_i\rho} \|\mathbf{y}_i^{t-1} - \mathbf{y}_i^*\|_2^2 - \frac{J\tau_i\rho}{2K} \|\mathbf{A}_i \mathbf{x}^t - \mathbf{a}_i\|_2^2 + \frac{(\gamma_i + \tau_i)\rho}{2} \|\mathbf{A}_i \mathbf{x}^t - \mathbf{a}_i\|_2^2 \right]$$

$$= \sum_{i=1}^{I} \left[ -\frac{K}{2J\tau_i\rho} \|\mathbf{y}_i^{t-1} - \mathbf{y}_i^*\|_2^2 + [\gamma_i + (1 - \frac{J}{K})\tau_i] \frac{\rho}{2} \|\mathbf{A}_i \mathbf{x}^t - \mathbf{a}_i\|_2^2 \right] . \tag{57}$$

$h(\mathbf{v}^*, \mathbf{v}^t)$ is reduced to

$$h(\mathbf{v}^*, \mathbf{v}^t) \geq \frac{\rho}{2} \sum_{i=1}^{I} [\gamma_i + (1 - \frac{J}{K})\tau_i] \|\mathbf{A}_i \mathbf{x}^t - \mathbf{a}_i\|_2^2 + \frac{\rho}{2} \|\mathbf{z}^* - \mathbf{z}^t\|_{\mathbf{Q}}^2 + \boldsymbol{\eta}^T B_\phi(\mathbf{x}^*, \mathbf{x}^t) . \tag{58}$$

Setting $1 - \nu_i = \frac{1}{\tilde{K}_i}$ and $\tau_i = \frac{K}{\tilde{K}_i(2J-K)}$, we have

$$\gamma_i + (1 - \frac{J}{K})\tau_i = (3 - \frac{2K}{J})(1 - \nu_i) + (1 - \frac{K}{J})\tau_i + \frac{1}{d_i} - \frac{2}{\tilde{K}_i} + (1 - \frac{J}{K})\tau_i$$

$$= (1 - \frac{K}{J})\frac{1}{\tilde{K}_i} + (2 - \frac{K}{J} - \frac{J}{K})\frac{K}{\tilde{K}_i(2J-K)} + \frac{1}{d_i} - \frac{K}{J\tilde{K}_i}$$

$$= \frac{(J-K)}{\tilde{K}_i(2J-K)} + \frac{1}{d_i} - \frac{K}{J\tilde{K}_i} \geq 0 . \tag{59}$$

Therefore, $h(\mathbf{v}^*, \mathbf{v}^t) \geq 0$. Letting $\zeta_i = \frac{J-K}{\tilde{K}_i(2J-K)} + \frac{1}{d_i} - \frac{K}{J\tilde{K}_i}$ completes the proof. ∎

The following theorem shows that $h(\mathbf{v}^*, \mathbf{v}^t)$ decreases monotonically and thus establishes the global convergence of PDMM.

**Theorem 1** *(Global Convergence of PDMM) Let* $\mathbf{v}^t = (\mathbf{x}_{j_t}^t, \mathbf{y}_i^t)$ *be generated by PDMM (3)-(5) and* $\mathbf{v}^* = (\mathbf{x}_j^*, \mathbf{y}_i^*)$ *be a KKT point satisfying (46)-(47). Setting* $\nu_i = 1 - \frac{1}{\tilde{K}_i}$ *and* $\tau_i = \frac{K}{\tilde{K}_i(2J-K)}$, *we have*

$$0 \leq \mathbb{E}_{\xi_t} h(\mathbf{v}^*, \mathbf{v}^{t+1}) \leq \mathbb{E}_{\xi_{t-1}} h(\mathbf{v}^*, \mathbf{v}^t) , \quad \mathbb{E}_{\xi_t} R(\mathbf{x}^{t+1}) \to 0 . \tag{60}$$

*Proof:* Adding (56) and (40) yields

$$0 \leq \sum_{i=1}^{I} \left\{ \langle \mathbf{y}_i^* - \mathbf{y}_i^t, \mathbf{A}_i^r \mathbf{x}^t - \mathbf{a}_i \rangle + \frac{\tau_i \rho}{2} \|\mathbf{A}_i^r \mathbf{x}^t - \mathbf{a}_i\|_2^2 \right\}$$

$$+ \frac{J}{K} \left\{ \tilde{\mathcal{L}}_\rho(\mathbf{x}^t, \mathbf{y}^t) - \mathbb{E}_{\mathbb{I}_t} \tilde{\mathcal{L}}_\rho(\mathbf{x}^{t+1}, \mathbf{y}^{t+1}) - \frac{\rho}{2} \sum_{i=1}^{I} \beta_i \mathbb{E}_{\mathbb{I}_t} \|\mathbf{A}_i^r \mathbf{x}^{t+1} - \mathbf{a}_i\|_2^2 \right.$$

$$+ \frac{\rho}{2} (\|\mathbf{z}^* - \mathbf{z}^t\|_{\mathbf{Q}}^2 - \mathbb{E}_{\mathbb{I}_t} \|\mathbf{z}^* - \mathbf{z}^{t+1}\|_{\mathbf{Q}}^2 - \mathbb{E}_{\mathbb{I}_t} \|\mathbf{z}^{t+1} - \mathbf{z}^t\|_{\mathbf{P}_t}^2)$$

$$\left. + \boldsymbol{\eta}^T (B_\phi(\mathbf{x}^*, \mathbf{x}^t) - \mathbb{E}_{\mathbb{I}_t} B_\phi(\mathbf{x}^*, \mathbf{x}^{t+1}) - \mathbb{E}_{\mathbb{I}_t} B_\phi(\mathbf{x}^{t+1}, \mathbf{x}^t)) \right\} . \tag{61}$$

Using (4), we have

$$\langle \mathbf{y}_i^* - \mathbf{y}_i^t, \mathbf{A}_i^r \mathbf{x}^t - \mathbf{a}_i \rangle + \frac{\tau_i \rho}{2} \|\mathbf{A}_i^r \mathbf{x}^t - \mathbf{a}_i\|_2^2 = \frac{1}{\tau_i \rho} \langle \mathbf{y}_i^* - \mathbf{y}_i^t, \mathbf{y}_i^t - \mathbf{y}_i^{t-1} \rangle + \frac{\tau_i \rho}{2} \|\mathbf{A}_i^r \mathbf{x}^t - \mathbf{a}_i\|_2^2$$

$$= \frac{1}{2\tau_i \rho} (\|\mathbf{y}_i^* - \mathbf{y}_i^{t-1}\|_2^2 - \|\mathbf{y}_i^* - \mathbf{y}_i^t\|_2^2) . \tag{62}$$

Plugging back into (61) gives

$$0 \leq \sum_{i=1}^{I} \frac{1}{2\tau_i \rho} (\|\mathbf{y}_i^* - \mathbf{y}_i^{t-1}\|_2^2 - \|\mathbf{y}_i^* - \mathbf{y}_i^t\|_2^2)$$

$$+ \frac{J}{K} \left\{ \tilde{\mathcal{L}}_\rho(\mathbf{x}^t, \mathbf{y}^t) - \mathbb{E}_{\mathbb{I}_t} \tilde{\mathcal{L}}_\rho(\mathbf{x}^{t+1}, \mathbf{y}^{t+1}) - \frac{\rho}{2} \sum_{i=1}^{I} \beta_i \mathbb{E}_{\mathbb{I}_t} \|\mathbf{A}_i^r \mathbf{x}^{t+1} - \mathbf{a}_i\|_2^2 \right.$$

$$+ \frac{\rho}{2} (\|\mathbf{z}^* - \mathbf{z}^t\|_{\mathbf{Q}}^2 - \mathbb{E}_{\mathbb{I}_t} \|\mathbf{z}^* - \mathbf{z}^{t+1}\|_{\mathbf{Q}}^2 - \mathbb{E}_{\mathbb{I}_t} \|\mathbf{z}^{t+1} - \mathbf{z}^t\|_{\mathbf{P}_t}^2)$$

$$\left. + \boldsymbol{\eta}^T (B_\phi(\mathbf{x}^*, \mathbf{x}^t) - \mathbb{E}_{\mathbb{I}_t} B_\phi(\mathbf{x}^*, \mathbf{x}^{t+1}) - \mathbb{E}_{\mathbb{I}_t} B_\phi(\mathbf{x}^{t+1}, \mathbf{x}^t)) \right\}$$

$$= \frac{J}{K} \left\{ h(\mathbf{v}^*, \mathbf{v}^t) - \mathbb{E}_{\mathbb{I}_t} h(\mathbf{v}^*, \mathbf{v}^{t+1}) - \mathbb{E}_{\mathbb{I}_t} R(\mathbf{x}^{t+1}) \right\} . \tag{63}$$

Taking expectaion over $\xi_{t-1}$, we have

$$0 \leq \frac{J}{K} \left\{ \mathbb{E}_{\xi_{t-1}} h(\mathbf{v}^*, \mathbf{v}^t) - \mathbb{E}_{\xi_t} h(\mathbf{v}^*, \mathbf{v}^{t+1}) - \mathbb{E}_{\xi_t} R(\mathbf{x}^{t+1}) \right\} . \tag{64}$$

Since $\mathbb{E}_{\xi_t} R(\mathbf{x}^{t+1}) \geq 0$, we have

$$\mathbb{E}_{\xi_t} h(\mathbf{v}^*, \mathbf{v}^{t+1}) \leq \mathbb{E}_{\xi_{t-1}} h(\mathbf{v}^*, \mathbf{v}^t) . \tag{65}$$

Thus, $\mathbb{E}_{\xi_t} h(\mathbf{v}^*, \mathbf{v}^{t+1})$ converges monotonically.

Rearranging the terms in (64) yields

$$\mathbb{E}_{\xi_t} R(\mathbf{x}^{t+1}) \leq \mathbb{E}_{\xi_{t-1}} h(\mathbf{v}^*, \mathbf{v}^t) - \mathbb{E}_{\xi_t} h(\mathbf{v}^*, \mathbf{v}^{t+1}) . \tag{66}$$

Summing over $t$ gives

$$\sum_{t=0}^{T-1} \mathbb{E}_{\xi_t} R(\mathbf{x}^{t+1}) \leq h(\mathbf{v}^*, \mathbf{v}^0) - \mathbb{E}_{\xi_{T-1}} h(\mathbf{v}^*, \mathbf{v}^T) \leq h(\mathbf{v}^*, \mathbf{v}^0) . \tag{67}$$

where the last inequality uses the Lemma 5. As $T \to \infty$, $\mathbb{E}_{\xi_t} R(\mathbf{x}^{t+1}) \to 0$, which completes the proof. ∎

The following theorem establishes the iteration complexity of PDMM in an ergodic sense.

**Theorem 2** *Let* $(\mathbf{x}_j^t, \mathbf{y}_i^t)$ *be generated by PDMM (3)-(5). Let* $\bar{\mathbf{x}}^T = \sum_{t=1}^{T} \mathbf{x}^t$. *Setting* $\nu_i = 1 - \frac{1}{\tilde{K}_i}$ *and* $\tau_i = \frac{K}{\tilde{K}_i(2J-K)}$, *we have*

$$\mathbb{E} f(\bar{\mathbf{x}}^T) - f(\mathbf{x}^*) \leq \frac{\sum_{i=1}^{I} \frac{1}{2\tau_i \rho} \|\mathbf{y}_i^0\|_2^2 + \frac{J}{K} \left\{ \frac{1}{2\beta_i \rho} \|\mathbf{y}_i^*\|_2^2 + \tilde{\mathcal{L}}_\rho(\mathbf{x}^1, \mathbf{y}^1) + \frac{\rho}{2} \|\mathbf{z}^* - \mathbf{z}^1\|_\mathbf{Q}^2 + \boldsymbol{\eta}^T B_\phi(\mathbf{x}^*, \mathbf{x}^1) \right\}}{T} ,$$
$$\tag{68}$$

$$\mathbb{E} \sum_{i=1}^{I} \beta_i \|\mathbf{A}_i^r \bar{\mathbf{x}}^T - \mathbf{a}_i\|_2^2 \leq \frac{\frac{2}{\rho} h(\mathbf{v}^*, \mathbf{v}^0)}{T} . \tag{69}$$

*where* $\beta_i = \frac{K}{J\tilde{K}_i}$.

*Proof:* Using (5) and, we have

$$-\sum_{i=1}^{I} \left\{ \langle \mathbf{y}_i^t, \mathbf{A}_i^r \mathbf{x}^t - \mathbf{a}_i \rangle - \frac{\tau_i \rho}{2} \|\mathbf{A}_i^r \mathbf{x}^t - \mathbf{a}_i\|_2^2 \right\}$$
$$= -\sum_{i=1}^{I} \left\{ \frac{1}{\tau_i \rho} \langle \mathbf{y}_i^t, \mathbf{y}_i^t - \mathbf{y}_i^{t-1} \rangle - \frac{1}{2\tau_i \rho} \|\mathbf{y}_i^t - \mathbf{y}_i^{t-1}\|_2^2 \right\}$$
$$= \sum_{i=1}^{I} \frac{1}{2\tau_i \rho} (\|\mathbf{y}_i^{t-1}\| - \|\mathbf{y}_i^t\|_2^2) . \tag{70}$$

Plugging back into (40) yields

$$f(\mathbf{x}^t) - f(\mathbf{x}^*) \leq \sum_{i=1}^{I} \frac{1}{2\tau_i \rho} (\|\mathbf{y}_i^{t-1}\|_2^2 - \|\mathbf{y}_i^t\|_2^2)$$
$$+ \frac{J}{K} \left\{ \tilde{\mathcal{L}}_\rho(\mathbf{x}^t, \mathbf{y}^t) - \mathbb{E}_{\mathbb{I}_t} \tilde{\mathcal{L}}_\rho(\mathbf{x}^{t+1}, \mathbf{y}^{t+1}) - \frac{\rho}{2} \sum_{i=1}^{I} \beta_i \mathbb{E}_{\mathbb{I}_t} \|\mathbf{A}_i^r \mathbf{x}^{t+1} - \mathbf{a}_i\|_2^2 \right.$$
$$+ \frac{\rho}{2} (\|\mathbf{z}^* - \mathbf{z}^t\|_\mathbf{Q}^2 - \mathbb{E}_{\mathbb{I}_t} \|\mathbf{z}^* - \mathbf{z}^{t+1}\|_\mathbf{Q}^2 - \mathbb{E}_{\mathbb{I}_t} \|\mathbf{z}^{t+1} - \mathbf{z}^t\|_{\mathbf{P}_t}^2)$$
$$\left. + \boldsymbol{\eta}^T (B_\phi(\mathbf{x}^*, \mathbf{x}^t) - \mathbb{E}_{\mathbb{I}_t} B_\phi(\mathbf{x}^*, \mathbf{x}^{t+1}) - \mathbb{E}_{\mathbb{I}_t} B_\phi(\mathbf{x}^{t+1}, \mathbf{x}^t)) \right\} . \tag{71}$$

Taking expectaion over $\xi_{t-1}$, we have

$$\mathbb{E}_{\xi_{t-1}} f(\mathbf{x}^t) - f(\mathbf{x}^*) \leq \sum_{i=1}^{I} \frac{1}{2\tau_i \rho} (\mathbb{E}_{\xi_{t-2}} \|\mathbf{y}_i^{t-1}\|_2^2 - \mathbb{E}_{\xi_{t-1}} \|\mathbf{y}_i^t\|_2^2)$$

$$
\begin{aligned}
+ \frac{J}{K} &\left\{ \mathbb{E}_{\xi_{t-1}} \tilde{\mathcal{L}}_\rho(\mathbf{x}^t, \mathbf{y}^t) - \mathbb{E}_{\xi_t} \tilde{\mathcal{L}}_\rho(\mathbf{x}^{t+1}, \mathbf{y}^{t+1}) - \frac{\rho}{2} \sum_{i=1}^I \beta_i \mathbb{E}_{\xi_t} \|\mathbf{A}_i^r \mathbf{x}^{t+1} - \mathbf{a}_i\|_2^2 \right. \\
&+ \frac{\rho}{2} (\mathbb{E}_{\xi_{t-1}} \|\mathbf{z}^* - \mathbf{z}^t\|_{\mathbf{Q}}^2 - \mathbb{E}_{\xi_t} \|\mathbf{z}^* - \mathbf{z}^{t+1}\|_{\mathbf{Q}}^2 - \mathbb{E}_{\xi_t} \|\mathbf{z}^{t+1} - \mathbf{z}^t\|_{\mathbf{P}_t}^2) \\
&\left. + \boldsymbol{\eta}^T (\mathbb{E}_{\xi_{t-1}} B_\phi(\mathbf{x}^*, \mathbf{x}^t) - \mathbb{E}_{\xi_t} B_\phi(\mathbf{x}^*, \mathbf{x}^{t+1}) - \mathbb{E}_{\xi_t} B_\phi(\mathbf{x}^{t+1}, \mathbf{x}^t)) \right\} .
\end{aligned} \tag{72}
$$

Summing over $t$, we have

$$
\begin{aligned}
\sum_{t=1}^T \mathbb{E}_{\xi_{t-1}} f(\mathbf{x}^t) - f(\mathbf{x}^*) \le & \sum_{i=1}^I \frac{1}{2\tau_i \rho} (\|\mathbf{y}_i^0\|_2^2 - \mathbb{E}_{\xi_{T-1}} \|\mathbf{y}_i^T\|_2^2) \\
&+ \frac{J}{K} \left\{ \tilde{\mathcal{L}}_\rho(\mathbf{x}^1, \mathbf{y}^1) - \mathbb{E}_{\xi_T} \tilde{\mathcal{L}}_\rho(\mathbf{x}^{T+1}, \mathbf{y}^{T+1}) - \frac{\rho}{2} \sum_{t=1}^T \sum_{i=1}^I \beta_i \mathbb{E}_{\xi_t} \|\mathbf{A}_i^r \mathbf{x}^{t+1} - \mathbf{a}_i\|_2^2 \right. \\
&+ \frac{\rho}{2} (\|\mathbf{z}^* - \mathbf{z}^1\|_{\mathbf{Q}}^2 - \mathbb{E}_{\xi_T} \|\mathbf{z}^* - \mathbf{z}^{T+1}\|_{\mathbf{Q}}^2 - \mathbb{E}_{\xi_T} \|\mathbf{z}^{T+1} - \mathbf{z}^T\|_{\mathbf{Q}}^2) \\
&\left. + \boldsymbol{\eta}^T (B_\phi(\mathbf{x}^*, \mathbf{x}^1) - \mathbb{E}_{\xi_T} B_\phi(\mathbf{x}^*, \mathbf{x}^{T+1}) - \mathbb{E}_{\xi_T} B_\phi(\mathbf{x}^{T+1}, \mathbf{x}^T)) \right\} .
\end{aligned} \tag{73}
$$

Using (56), we have

$$
\begin{aligned}
\tilde{\mathcal{L}}_\rho(\mathbf{x}^{T+1}, \mathbf{y}^{T+1}) &= f(\mathbf{x}^{T+1}) - f(\mathbf{x}^*) + \sum_{i=1}^I [\langle \mathbf{y}_i^{T+1}, \mathbf{A}_i \mathbf{x}^{T+1} - \mathbf{a}_i \rangle + \frac{(\gamma_i - \tau_i)\rho}{2} \|\mathbf{A}_i \mathbf{x}^{T+1} - \mathbf{a}_i\|_2^2] \\
&\ge - \sum_{i=1}^I \langle \mathbf{y}_i^*, \mathbf{A}_i^r \mathbf{x}^{T+1} - \mathbf{a}_i \rangle + \sum_{i=1}^I [\langle \mathbf{y}_i^T, \mathbf{A}_i \mathbf{x}^{T+1} - \mathbf{a}_i \rangle + \frac{(\gamma_i + \tau_i)\rho}{2} \|\mathbf{A}_i \mathbf{x}^{T+1} - \mathbf{a}_i\|_2^2] \\
&\ge - \sum_{i=1}^I (\frac{1}{2\delta_i} \|\mathbf{y}_i^*\|_2^2 + \frac{\delta_i}{2} \|\mathbf{A}_i^r \mathbf{x}^{T+1} - \mathbf{a}_i\|_2^2) + \sum_{i=1}^I \left[ -\frac{K}{2J\tau_i\rho} \|\mathbf{y}_i^T\|_2^2 + [\gamma_i + (1 - \frac{J}{K})\tau_i] \frac{\rho}{2} \|\mathbf{A}_i \mathbf{x}^{T+1} - \mathbf{a}_i\|_2^2 \right] \\
&\ge - \sum_{i=1}^I (\frac{1}{2\delta_i} \|\mathbf{y}_i^*\|_2^2 + \frac{\delta_i}{2} \|\mathbf{A}_i^r \mathbf{x}^{T+1} - \mathbf{a}_i\|_2^2) - \sum_{i=1}^I \frac{K}{2J\tau_i\rho} \|\mathbf{y}_i^T\|_2^2 ,
\end{aligned} \tag{74}
$$

where $\delta_i > 0$ and the last inequality uses (59). Plugging into (73), we have

$$
\begin{aligned}
\sum_{t=1}^T \mathbb{E}_{\xi_{t-1}} f(\mathbf{x}^t) - f(\mathbf{x}^*) \le & \sum_{i=1}^I \frac{1}{2\tau_i\rho} \|\mathbf{y}_i^0\|_2^2 + \frac{J}{K} \left\{ \tilde{\mathcal{L}}_\rho(\mathbf{x}^1, \mathbf{y}^1) + \frac{\rho}{2} \|\mathbf{z}^* - \mathbf{z}^1\|_{\mathbf{Q}}^2 + \boldsymbol{\eta}^T B_\phi(\mathbf{x}^*, \mathbf{x}^1) \right\} \\
&+ \frac{J}{K} \left\{ \sum_{i=1}^I \left[ \frac{1}{2\delta_i} \|\mathbf{y}_i^*\|_2^2 + \frac{\delta_i - \beta_i\rho}{2} \mathbb{E} \|\mathbf{A}_i^r \mathbf{x}^{T+1} - \mathbf{a}_i\|_2^2 \right] \right\} .
\end{aligned} \tag{75}
$$

Settin $\delta_i = \beta_i \rho$, dividing by $T$ and letting $\bar{\mathbf{x}}^T = \frac{1}{T} \sum_{t=1}^T \mathbf{x}^t$ complete the proof.

Dividing both sides of (67) by $T$ yields (69). ∎

# 2 Connection to ADMM

In this section, we use ADMM to solve (1), similar as [5, 3] but with different forms. We show that ADMM is a speical case of PDMM. The connection can help us understand why the two parameters $\tau_i, \nu_i$ in PDMM are necessary. We first introduce splitting variables $\mathbf{z}_i$ as follows:

$$\min \sum_{j=1}^{J} f_j(\mathbf{x}_j) \quad \text{s.t.} \quad \mathbf{A}_j\mathbf{x}_j = \mathbf{z}_j, \sum_{j=1}^{J} \mathbf{z}_j = \mathbf{a} , \tag{76}$$

which can be written as

$$\min \sum_{j=1}^{K} f_j(\mathbf{x}_j) + g(\mathbf{z}) \quad \text{s.t.} \quad \mathbf{A}_j\mathbf{x}_j = \mathbf{z}_j , \tag{77}$$

where $g(\mathbf{z})$ is an indicator function of $\sum_{j=1}^{K} \mathbf{z}_j = \mathbf{a}$. The augmented Lagrangian is

$$\mathcal{L}_\rho(\mathbf{x}_j, \mathbf{z}_j, \mathbf{y}_j) = \sum_{j=1}^{J} \left[ f_j(\mathbf{x}_j) + \langle \mathbf{y}_j, \mathbf{A}_j\mathbf{x}_j - \mathbf{z}_j \rangle + \frac{\rho}{2}\|\mathbf{A}_j\mathbf{x}_j - \mathbf{z}_j\|_2^2 \right] , \tag{78}$$

where $\mathbf{y}_j$ is the dual variable. We have the following ADMM iterates:

$$\mathbf{x}_j^{t+1} = \text{argmin}_{\mathbf{x}_i} \ f_j(\mathbf{x}_j) + \langle \mathbf{y}_j^t, \mathbf{A}_j\mathbf{x}_j - \mathbf{z}_j^t \rangle + \frac{\rho}{2}\|\mathbf{A}_j\mathbf{x}_j - \mathbf{z}_j^t\|_2^2 , \tag{79}$$

$$\mathbf{z}^{t+1} = \text{argmin}_{\sum_{j=1}^{K} \mathbf{z}_j = \mathbf{a}} \sum_{j=1}^{K} \left[ \langle \mathbf{y}_i^t, \mathbf{A}_j\mathbf{x}_j^{t+1} - \mathbf{z}_j \rangle + \frac{\rho}{2}\|\mathbf{A}_j\mathbf{x}_j^{t+1} - \mathbf{z}_j\|_2^2 \right] , \tag{80}$$

$$\mathbf{y}_j^{t+1} = \mathbf{y}_j^t + \rho(\mathbf{A}_j\mathbf{x}_j^{t+1} - \mathbf{z}_j^{t+1}) . \tag{81}$$

The Lagrangian of (80) is

$$\mathcal{L} = \sum_{j=1}^{J} \left[ \langle \mathbf{y}_j^t, \mathbf{A}_j\mathbf{x}_j^{t+1} - \mathbf{z}_j \rangle + \frac{\rho}{2}\|\mathbf{A}_j\mathbf{x}_j^{t+1} - \mathbf{z}_j\|_2^2 \right] + \langle \boldsymbol{\lambda}, \sum_{j=1}^{J} \mathbf{z}_j - \mathbf{a} \rangle , \tag{82}$$

where $\boldsymbol{\lambda}$ is the dual variable. The first order optimality is

$$-\mathbf{y}_j^t + \rho(\mathbf{z}_j^{t+1} - \mathbf{A}_j\mathbf{x}_j^{t+1}) + \boldsymbol{\lambda} = 0 . \tag{83}$$

Using (81) gives

$$\boldsymbol{\lambda} = \mathbf{y}_j^{t+1}, \quad \forall j . \tag{84}$$

Denoting $\mathbf{y}^t = \mathbf{y}_j^t$, (83) becomes

$$\mathbf{y}^{t+1} = \mathbf{y}^t + \rho(\mathbf{A}_j\mathbf{x}_j^{t+1} - \mathbf{z}_j^{t+1}) . \tag{85}$$

Summing over $j$ and using the constraint $\sum_{j=1}^{J} \mathbf{z}_i = \mathbf{a}$, we have

$$\mathbf{y}^{t+1} = \mathbf{y}^t + \frac{\rho}{J}(\mathbf{A}\mathbf{x}^{t+1} - \mathbf{a}) . \tag{86}$$

Subtracting (85) from (86), simple calculations yields

$$\mathbf{z}_j^{t+1} = \mathbf{A}_j \mathbf{x}_j^{t+1} + \frac{1}{J}(\mathbf{A}\mathbf{x}^{t+1} - \mathbf{a}) . \tag{87}$$

Plugging into (79), we have

$$
\begin{aligned}
\mathbf{x}_j^{t+1} &= \operatorname{argmin}_{\mathbf{x}_j} \ f_j(\mathbf{x}_j) + \langle \mathbf{y}^t, \mathbf{A}_j \mathbf{x}_j \rangle + \frac{\rho}{2} \|\mathbf{A}_j \mathbf{x}_j - \mathbf{z}_j^t\|_2^2 \\
&= \operatorname{argmin}_{\mathbf{x}_j} \ f_j(\mathbf{x}_j) + \langle \mathbf{y}^t, \mathbf{A}_j \mathbf{x}_j \rangle + \frac{\rho}{2} \|\mathbf{A}_j \mathbf{x}_j - \mathbf{A}_j \mathbf{x}_j^t + \frac{\mathbf{A}\mathbf{x}^t - \mathbf{a}}{J}\|_2^2 \\
&= \operatorname{argmin}_{\mathbf{x}_j} \ f_j(\mathbf{x}_j) + \langle \hat{\mathbf{y}}^t, \mathbf{A}_j \mathbf{x}_j \rangle + \frac{\rho}{2} \|\mathbf{A}_j \mathbf{x}_j + \sum_{k \neq j} \mathbf{A}_k \mathbf{x}_k^t - \mathbf{a}\|_2^2 ,
\end{aligned}
\tag{88}
$$

where $\hat{\mathbf{y}}^t = \mathbf{y}^t - (1 - \frac{1}{J})\rho(\mathbf{A}\mathbf{x}^t - \mathbf{a})$, which becomes PDMM by setting $\tau = \frac{1}{J}, \nu = 1 - \frac{1}{J}$ and updating all blocks. Therefore, splitting ADMM (SADMM) is a special case of PDMM.

# 3 Inexact PDMM and connection to PJADMM

In this section, we only consider the case when all blocks are used in PDMM. We show that if setting $\eta_j$ sufficiently large, the dual backward step (5) is not needed, which becomes PJADMM [2]. Together with the connection between PDMM and sADMM in Section 2, sADMM and PJADMM are two extreme cases of PDMM. If the primal update makes sufficient progress, the dual update should take small step, e.g., sADMM. On the other hand, if the primal update makes conservative progress, the dual update can take full gradient step, e.g. PJADMM. While sADMM is a direct derivation of ADMM, PJADMM introduces more terms and parameters.

**Corollary 1** *Let $\{\mathbf{x}_j^t, \mathbf{y}_i^t\}$ be generated by PDMM (3)-(5). Assume $\tau_i > 0$ and $\nu_i \geq 0$. We have*

$$
\begin{aligned}
f(\mathbf{x}^{t+1}) - f(\mathbf{x}^*) \leq {}& \sum_{i=1}^{I} \left\{ -\langle \mathbf{y}_i^{t+1}, \mathbf{A}_i^r \mathbf{x}^{t+1} - \mathbf{a}_i \rangle + \frac{\tau_i \rho}{2} \|\mathbf{A}_i^r \mathbf{x}^{t+1} - \mathbf{a}_i\|_2^2 \right\} \\
& + \frac{\rho}{2}(\|\mathbf{z}^t - \mathbf{z}^*\|_\mathbf{Q}^2 - \|\mathbf{z}^{t+1} - \mathbf{z}^*\|_\mathbf{Q}^2 - \|\mathbf{z}^{t+1} - \mathbf{z}^t\|_\mathbf{Q}^2) \\
& + \frac{\rho}{2} \sum_{i=1}^{I} \Bigg\{ (\nu_i - 1 + \frac{1}{d_i})(\|\mathbf{A}_i^r \mathbf{x}^t - \mathbf{a}_i\|_2^2 - \|\mathbf{A}_i^r \mathbf{x}^{t+1} - \mathbf{a}_i\|_2^2) \\
& + (\tau_i + 2\nu_i - 2)\|\mathbf{A}_i^r \mathbf{x}^{t+1} - \mathbf{a}_i\|_2^2 + (1 - \nu_i - \frac{1}{d_i})\|\mathbf{A}_i^r(\mathbf{x}^{t+1} - \mathbf{x}^t)\|_2^2 \Bigg\} \\
& + \sum_{j=1}^{J} \eta_j \left( B_{\phi_j}(\mathbf{x}_j^*, \mathbf{x}_j^t) - B_{\phi_j}(\mathbf{x}_j^*, \mathbf{x}_j^{t+1}) - B_{\phi_j}(\mathbf{x}_j^{t+1}, \mathbf{x}_j^t) \right) .
\end{aligned}
\tag{89}
$$

*Proof:* Let $\mathbb{I}_t$ be all blocks, $K = J$. According the definition of $\mathbf{P}_t$ in (7) and $\mathbf{Q}$ in (13), $\mathbf{P}_t = \mathbf{Q}$. Therefore, (16) reduces to

$$f(\mathbf{x}^{t+1}) - f(\mathbf{x}^*) \leq \sum_{i=1}^{I} \left\{ -\langle \mathbf{y}_i^{t+1}, \mathbf{A}_i^r \mathbf{x}^{t+1} - \mathbf{a}_i \rangle + \frac{\tau_i \rho}{2} \|\mathbf{A}_i^r \mathbf{x}^{t+1} - \mathbf{a}_i\|_2^2 \right\}$$

$$+ \frac{\rho}{2}(\|\mathbf{z}^t - \mathbf{z}^*\|_{\mathbf{Q}}^2 - \|\mathbf{z}^{t+1} - \mathbf{z}^*\|_{\mathbf{Q}}^2 - \|\mathbf{z}^{t+1} - \mathbf{z}^t\|_{\mathbf{Q}}^2)$$

$$+ \sum_{j=1}^{J} \eta_j \left( B_{\phi_j}(\mathbf{x}_j^*, \mathbf{x}_j^t) - B_{\phi_j}(\mathbf{x}_j^*, \mathbf{x}_j^{t+1}) - B_{\phi_j}(\mathbf{x}_j^{t+1}, \mathbf{x}_j^t) \right)$$

$$+ \frac{\rho}{2} \sum_{i=1}^{I} \left\{ (\nu_i - 1 + \frac{1}{d_i})\|\mathbf{A}_i^r \mathbf{x}^t - \mathbf{a}_i\|_2^2 - (1 - \nu_i - \tau_i + \frac{1}{d_i})\|\mathbf{A}_i^r \mathbf{x}^{t+1} - \mathbf{a}_i\|_2^2 + (1 - \nu_i - \frac{1}{d_i})\|\mathbf{A}_i^r(\mathbf{x}^{t+1} - \mathbf{x}^t)\|_2^2 \right\} .$$

$$(90)$$

Rearranging the terms completes the proof. ∎

**Corollary 2** *Let $\{\mathbf{x}_j^t, \mathbf{y}_i^t\}$ be generated by PDMM (3)-(5). Assume (1)$\tau_i > 0$ and $\nu_i \geq 0$; (2) $\eta_j > 0$; (3) $\phi_j$ is $\alpha_j$-strongly convex. We have*

$$f(\mathbf{x}^{t+1}) - f(\mathbf{x}^*) \leq \sum_{i=1}^{I} \left\{ -\langle \mathbf{y}_i^{t+1}, \mathbf{A}_i^r \mathbf{x}^{t+1} - \mathbf{a}_i \rangle + \frac{\tau_i \rho}{2} \|\mathbf{A}_i^r \mathbf{x}^{t+1} - \mathbf{a}_i\|_2^2 \right\}$$

$$+ \frac{\rho}{2}(\|\mathbf{z}^t - \mathbf{z}^*\|_{\mathbf{Q}}^2 - \|\mathbf{z}^{t+1} - \mathbf{z}^*\|_{\mathbf{Q}}^2 - \|\mathbf{z}^{t+1} - \mathbf{z}^t\|_{\mathbf{Q}}^2)$$

$$+ \sum_{j=1}^{J} \eta_j \left( B_{\phi_j}(\mathbf{x}_j^*, \mathbf{x}_j^t) - B_{\phi_j}(\mathbf{x}_j^*, \mathbf{x}_j^{t+1}) \right) . \tag{91}$$

*$\nu_i$ and $\tau_i$ satisfy $\nu_i \in [1 - \frac{1}{d_i} - \frac{\eta_j \alpha_j}{\rho I d_i \lambda_{\max}^{ij}}, 1 - \frac{1}{d_i}]$ and $\tau_i \leq 1 + \frac{1}{d_i} - \nu_i$, where $\lambda_{\max}^{ij}$ is the largest eigenvalue of $\mathbf{A}_{ij}^T \mathbf{A}_{ij}$. In particular, if $\eta_j = \frac{(d_i-1)\rho I \lambda_{\max}^{ij}}{\alpha_j}$, $\nu_i = 0$ and $\tau_i \leq 1 + \frac{1}{d_i}$.*

*Proof:*   Assume $\eta_j > 0$. We can choose larger $\tau_i$ and smaller $\nu_i$ than Lemma 3 by setting $\eta_j$ sufficiently large. Since $\phi_j$ is $\alpha_j$-strongly convex, $B_{\phi_j}(\mathbf{x}_j^{t+1}, \mathbf{x}_j^t) \geq \frac{\alpha_j}{2}\|\mathbf{x}_j^{t+1} - \mathbf{x}_j^t\|_2^2$. We have

$$\sum_{j=1}^{J} \eta_j B_{\phi_j}(\mathbf{x}_j^{t+1}, \mathbf{x}_j^t) \geq \sum_{i=1}^{I} \sum_{j=1}^{J} \frac{\eta_j \alpha_j}{2I} \|\mathbf{x}_j^{t+1} - \mathbf{x}_j^t\|_2^2 \geq \sum_{i=1}^{I} \sum_{j \in \mathcal{N}(i)} \frac{\eta_j \alpha_j}{2I \lambda_{\max}^{ij}} \|\mathbf{A}_{ij}(\mathbf{x}_j^{t+1} - \mathbf{x}_j^t)\|_2^2 . \tag{92}$$

$$\|\mathbf{A}_i^r(\mathbf{x}^{t+1} - \mathbf{x}^t)\|_2^2 = \| \sum_{j \in \mathcal{N}(i)} \mathbf{A}_{ij}(\mathbf{x}_j^{t+1} - \mathbf{x}_j^t)\|_2^2 \leq d_i \sum_{j \in \mathcal{N}(i)} \|\mathbf{A}_{ij}(\mathbf{x}_j^{t+1} - \mathbf{x}_j^t)\|_2^2 , \tag{93}$$

where $\lambda_{\max}^{ij}$ is the largest eigenvalue of $\mathbf{A}_{ij}^T \mathbf{A}_{ij}$. Plugging into (89) gives

$$f(\mathbf{x}^{t+1}) - f(\mathbf{x}^*) \leq \sum_{i=1}^{I} \left\{ -\langle \mathbf{y}_i^{t+1}, \mathbf{A}_i^r \mathbf{x}^{t+1} - \mathbf{a}_i \rangle + \frac{\tau_i \rho}{2} \|\mathbf{A}_i^r \mathbf{x}^{t+1} - \mathbf{a}_i\|_2^2 \right\}$$

$$+ \frac{\rho}{2}(\|\mathbf{z}^t - \mathbf{z}^*\|_{\mathbf{Q}}^2 - \|\mathbf{z}^{t+1} - \mathbf{z}^*\|_{\mathbf{Q}}^2 - \|\mathbf{z}^{t+1} - \mathbf{z}^t\|_{\mathbf{Q}}^2)$$

$$+ \frac{\rho}{2} \sum_{i=1}^{I} \left\{ (\nu_i - 1 + \frac{1}{d_i})(\|\mathbf{A}_i^r \mathbf{x}^t - \mathbf{a}_i\|_2^2 - \|\mathbf{A}_i^r \mathbf{x}^{t+1} - \mathbf{a}_i\|_2^2) \right\}$$

$$+(\tau_i + 2\nu_i - 2)\|\mathbf{A}_i^r \mathbf{x}^{t+1} - \mathbf{a}_i\|_2^2 + \sum_{j \in \mathcal{N}(i)} [(1 - \nu_i)d_i - 1 - \frac{\eta_j \alpha_j}{\rho I \lambda_{\max}^{ij}}]\|\mathbf{A}_{ij}(\mathbf{x}_j^{t+1} - \mathbf{x}_j^t)\|_2^2 \Big\}$$

$$+ \sum_{j=1}^{J} \eta_j \left( B_{\phi_j}(\mathbf{x}_j^*, \mathbf{x}_j^t) - B_{\phi_j}(\mathbf{x}_j^*, \mathbf{x}_j^{t+1}) \right) . \tag{94}$$

If $(1 - \nu_i)d_i - 1 - \frac{\eta_j \alpha_j}{\rho I \lambda_{\max}^{ij}} \leq 0$, i.e., $\nu_i \geq 1 - \frac{1}{d_i} - \frac{\eta_j \alpha_j}{\rho I d_i \lambda_{\max}^{ij}}$, we have

$$f(\mathbf{x}^{t+1}) - f(\mathbf{x}^*) \leq \frac{\rho}{2} \sum_{i=1}^{I} \left\{ -\frac{2}{\rho} \langle \mathbf{y}_i^{t+1}, \mathbf{A}_i^r \mathbf{x}^{t+1} - \mathbf{a}_i \rangle + \tau_i \|\mathbf{A}_i^r \mathbf{x}^{t+1} - \mathbf{a}_i\|_2^2 \right\}$$

$$+ \frac{\rho}{2}(\|\mathbf{z}^t - \mathbf{z}^*\|_\mathbf{Q}^2 - \|\mathbf{z}^{t+1} - \mathbf{z}^*\|_\mathbf{Q}^2 - \|\mathbf{z}^{t+1} - \mathbf{z}^t\|_\mathbf{Q}^2)$$

$$+ \sum_{i=1}^{J} \eta_i \left( B_{\phi_i}(\mathbf{x}_j^*, \mathbf{x}_j^t) - B_{\phi_i}(\mathbf{x}_j^*, \mathbf{x}_j^{t+1}) \right)$$

$$+ \frac{\rho}{2} \sum_{i=1}^{I} \left\{ -(\nu_i - 1 + \frac{1}{d_i})\|\mathbf{A}_i^r \mathbf{x}^{t+1} - \mathbf{a}_i\|_2^2 + (\tau_i - 2 + 2\nu_i)\|\mathbf{A}_i^r \mathbf{x}^{t+1} - \mathbf{a}_i\|_2^2 \right\} . \tag{95}$$

If $\tau_i - 2 + 2\nu_i - (\nu_i - 1 + \frac{1}{d_i}) \leq 0$, i.e., $\tau_i \leq 1 + \frac{1}{d_i} - \nu_i$, the last two terms in (95) can be removed. Therefore, when $\nu_i \geq 1 - \frac{1}{d_i} - \frac{\eta_j \alpha_j}{\rho I d_i \lambda_{\max}^{ij}}$ and $\tau_i \leq 1 + \frac{1}{d_i} - \nu_i$, we have (91). $\blacksquare$

Define the current iterate $\mathbf{v}^t = (\mathbf{x}_j^t, \mathbf{y}_i^t)$ and $h(\mathbf{v}^*, \mathbf{v}^t)$ as a distance from $\mathbf{v}^t$ to a KKT point $\mathbf{v}^* = (\mathbf{x}_j^*, \mathbf{y}_i^*)$:

$$h(\mathbf{v}^*, \mathbf{v}^t) = \sum_{i=1}^{I} \frac{1}{2\tau_i \rho} \|\mathbf{y}_i^* - \mathbf{y}_i^t\|_2^2 + \frac{\rho}{2} \|\mathbf{u}^t - \mathbf{u}^*\|_\mathbf{Q}^2 + \sum_{j=1}^{J} \eta_j B_{\phi_j}(\mathbf{x}_j^*, \mathbf{x}_j^t) . \tag{96}$$

The following theorem shows that $h(\mathbf{v}^*, \mathbf{v}^t)$ decreases monotonically and thus establishes the global convergence of PDMM.

**Theorem 3** *(Global Convergence of PDMM) Let $\mathbf{v}^t = (\mathbf{x}_j^t, \mathbf{y}_i^t)$ be generated by PDMM (3)-(5) and $\mathbf{v}^* = (\mathbf{x}_j^*, \mathbf{y}_i^*)$ be a KKT point satisfying (46)-(47). Assume $\tau_i, \nu_i$ and $\gamma_i$ satisfy conditions in Lemma 2. Then $\mathbf{v}^t$ converges to the KKT point $\mathbf{v}^*$ monotonically, i.e.,*

$$h(\mathbf{v}^*, \mathbf{v}^{t+1}) \leq h(\mathbf{v}^*, \mathbf{v}^t) \tag{97}$$

*Proof:* Adding (56) and (91) together yields

$$0 \leq \sum_{i=1}^{I} \left\{ \langle \mathbf{y}_i^* - \mathbf{y}_i^{t+1}, \mathbf{A}_i^r \mathbf{x}^{t+1} - \mathbf{a}_i \rangle + \frac{\tau_i \rho}{2} \|\mathbf{A}_i^r \mathbf{x}^{t+1} - \mathbf{a}_i\|_2^2 \right\}$$

$$+ \frac{\rho}{2}(\|\mathbf{u}^t - \mathbf{u}^*\|_\mathbf{Q}^2 - \|\mathbf{u}^{t+1} - \mathbf{u}^*\|_\mathbf{Q}^2 - \|\mathbf{u}^{t+1} - \mathbf{u}^t\|_\mathbf{Q}^2)$$

$$+ \sum_{j=1}^{J} \eta_j \left( B_{\phi_j}(\mathbf{x}_j^*, \mathbf{x}_j^t) - B_{\phi_j}(\mathbf{x}_j^*, \mathbf{x}_j^{t+1}) \right) . \tag{98}$$

The first term in the bracket can be rewritten as

$$\langle \mathbf{y}_i^* - \mathbf{y}_i^{t+1}, \mathbf{A}_i^r \mathbf{x}^{t+1} - \mathbf{a}_i \rangle = \frac{1}{\tau_i \rho} \langle \mathbf{y}_i^* - \mathbf{y}_i^{t+1}, \mathbf{y}_i^{t+1} - \mathbf{y}_i^t \rangle$$

$$= \frac{1}{2\tau_i \rho} \left( \|\mathbf{y}_i^* - \mathbf{y}_i^t\|_2^2 - \|\mathbf{y}_i^* - \mathbf{y}_i^{t+1}\|_2^2 - \|\mathbf{y}_i^{t+1} - \mathbf{y}_i^t\|_2^2 \right)$$

$$= \frac{1}{2\tau_i \rho} \left( \|\mathbf{y}_i^* - \mathbf{y}_i^t\|_2^2 - \|\mathbf{y}_i^* - \mathbf{y}_i^{t+1}\|_2^2 \right) - \frac{\tau_i \rho}{2} \|\mathbf{A}_i^r \mathbf{x}^{t+1} - \mathbf{a}_i\|_2^2 . \tag{99}$$

Plugging back into (98) yields

$$0 \le \sum_{i=1}^{I} \frac{1}{2\tau_i \rho} \left( \|\mathbf{y}_i^* - \mathbf{y}_i^t\|_2^2 - \|\mathbf{y}_i^* - \mathbf{y}_i^{t+1}\|_2^2 \right)$$

$$+ \frac{\rho}{2} (\|\mathbf{u}^t - \mathbf{u}^*\|_{\mathbf{Q}}^2 - \|\mathbf{u}^{t+1} - \mathbf{u}^*\|_{\mathbf{Q}}^2 - \|\mathbf{u}^{t+1} - \mathbf{u}^t\|_{\mathbf{Q}}^2)$$

$$+ \sum_{j=1}^{J} \eta_j \left( B_{\phi_j}(\mathbf{x}_j^*, \mathbf{x}_j^t) - B_{\phi_j}(\mathbf{x}_j^*, \mathbf{x}_j^{t+1}) \right) . \tag{100}$$

Rearranging the terms completes the proof. ∎

The following theorem establishes the $O(1/T)$ convergence rate for the objective in an ergodic sense.

**Theorem 4** *Let $(\mathbf{x}_j^t, \mathbf{y}_i^t)$ be generated by PDMM (3)-(5). Assume $\tau_i, \nu_i \ge 0$ satisfy conditions in Lemma 2. Let $\bar{\mathbf{x}}^T = \sum_{t=1}^{T} \mathbf{x}^t$. We have*

$$f(\bar{\mathbf{x}}^T) - f(\mathbf{x}^*) \le \frac{\frac{1}{2\tau\rho} \|\mathbf{y}^0\|_2^2 + \frac{\rho}{2} \|\mathbf{u}^0 - \mathbf{u}^*\|_{\mathbf{Q}}^2 + \sum_{j=1}^{J} \eta_j B_{\phi_j}(\mathbf{x}_j^*, \mathbf{x}_j^0)}{T} , \tag{101}$$

*Proof:* Using (4), we have

$$- \langle \mathbf{y}_i^{t+1}, \mathbf{A}_i^r \mathbf{x}^{t+1} - \mathbf{a}_i \rangle = -\frac{1}{\tau_i \rho} \langle \mathbf{y}_i^{t+1}, \mathbf{y}_i^{t+1} - \mathbf{y}_i^t \rangle$$

$$= \frac{1}{2\tau_i \rho} (\|\mathbf{y}_i^t\|_2^2 - \|\mathbf{y}_i^{t+1}\|_2^2 - \|\mathbf{y}_i^{t+1} - \mathbf{y}_i^t\|_2^2)$$

$$= \frac{1}{2\tau_i \rho} (\|\mathbf{y}_i^t\|_2^2 - \|\mathbf{y}_i^{t+1}\|_2^2) - \frac{\tau_i \rho}{2} \|\mathbf{A}_i^r \mathbf{x}^{t+1} - \mathbf{a}_i\|_2^2 . \tag{102}$$

Plugging into (91) yields

$$f(\mathbf{x}^{t+1}) - f(\mathbf{x}^*) \le \sum_{i=1}^{I} \frac{1}{2\tau_i \rho} (\|\mathbf{y}_i^t\|_2^2 - \|\mathbf{y}_i^{t+1}\|_2^2)$$

$$+ \frac{\rho}{2} (\|\mathbf{u}^t - \mathbf{u}^*\|_{\mathbf{Q}}^2 - \|\mathbf{u}^{t+1} - \mathbf{u}^*\|_{\mathbf{Q}}^2 - \|\mathbf{u}^{t+1} - \mathbf{u}^t\|_{\mathbf{Q}}^2)$$

$$+ \sum_{j=1}^{J} \eta_j \left( B_{\phi_j}(\mathbf{x}_j^*, \mathbf{x}_j^t) - B_{\phi_j}(\mathbf{x}_j^*, \mathbf{x}_j^{t+1}) \right) . \tag{103}$$

Summing over $t$ from $0$ to $T-1$, we have

$$\sum_{t=0}^{T-1} \left[ f(\mathbf{x}^{t+1}) - f(\mathbf{x}^*) \right] \leq \sum_{i=1}^{I} \frac{1}{2\tau_i \rho}(\|\mathbf{y}_i^t\|_2^2 - \|\mathbf{y}_i^{t+1}\|_2^2)$$
$$+ \frac{\rho}{2}(\|\mathbf{u}^0 - \mathbf{u}^*\|_{\mathbf{Q}}^2 - \|\mathbf{u}^T - \mathbf{u}^*\|_{\mathbf{Q}}^2)$$
$$+ \sum_{j=1}^{J} \eta_j \left( B_{\phi_j}(\mathbf{x}_j^*, \mathbf{x}_j^t) - B_{\phi_j}(\mathbf{x}_j^*, \mathbf{x}_j^{t+1}) \right) . \tag{104}$$

Applying the Jensen's inequality on the LHS and using $\bar{\mathbf{x}}^T = \sum_{t=1}^{T} \mathbf{x}^t$ complete the proof. ∎

If $\eta_j = \frac{(d_i-1)\rho I \lambda_{\max}^{ij}}{\alpha_j}$, $\nu_i = 0$ and $\tau_i = 1$. Therefore, PDMM becomes PJADMM [2], where the convergence rate of PJADMM has been improved to $o(1/T)$.