[Reviews · NeurIPS 2014]

Submitted by Assigned_Reviewer_16

This paper considers the problem of minimizing block-separable functions subject to linear constraints. A variant of ADMM (called PDMM) is proposed to deal with the multiple blocks case in this paper: it randomly selects K blocks of the primal variable to update and uses a backward step to refine the dual variable; the process to update the primal variable can be implemented in parallel. The sublinear convergence rate is proved for the proposed method. This method generalized several variants of ADMM.

This paper is well written and easy to follow. I am positive to accept this paper. My questions and comments are given below:

1) The algorithm description in (5) and (6) is not clear enough. Do you update all coordinates of y^{t+1} and \hat{y}^{t+1} or just a single block "i"? My understanding is that you only update a block of the dual variable "y", but I did not find how to select "i". This should be clarified in the revision.

2) In Theorem 2, I believe that "\sum_{i=1}^I" was missing behind of "{" from your proof.

3) More discussion about Theorem should be included. Theorem 2 (the main result in this paper) provides the convergence rate of the proposed method PDMM. First, PDMM generalizes several variants of ADMM. A natural question is if Theorem 2 (by properly choose parameters like K, J) is consistent with the convergence rates of those variants. Second, since you split the transformation matrix A into multiple blocks, what is the optimal way to split it from your theorem?

4) I am curious of the comparison on the tensor completion problem ("tensor completion for estimating missing values in visual data, 2012"), which also has the multiple block structure. Do you have any clue which variant of ADMM is optimal?

Minors:

1) In (9), a space is missing behind of "min"

2) Line 217, remove the space before "Section"
Summary: I am positive to accept this paper. More discussion after Theorem 2 is expected.

Submitted by Assigned_Reviewer_19

This paper proposes a parallel direction method of multipliers (PDMM) to minimize block-separable convex functions subject to linear constraints. In contrast with ADMM, which update primal variables using a Gauss-Seidel manner but limited to two blocks, this work proposes to update the primal blocks in the Jacobian manner with multiple blocks. The main result of the paper is the introduction of the dual backward step, which compensate the limited information propagation in the Jacobian updates by effectively reducing the step size for the dual updates. Similar to ADMM, O(1/T) type convergence results are established for PDMM in this paper.

It is shown (in supplementary materials) that two previous methods, sADMM and PJADMM are special cases of the proposed method, which give better understanding the relationship between different methods in the landscape of decomposition methods based on augmented Lagrangian. Experiments results demonstrated the effectiveness of the PDMM method as compared with ADMM and other variants. The results are a little counter intuitive for me, since in general Jacobian type methods would give slow convergence compared with Gauss-Seidel type of methods. It would be good to elaborate on this point.
Summary: The dual backward step in PDMM to ensure convergence seem to be new and key to obtain convergence under general conditions as ADMM. The proposed method gives more alternatives and possibilities for distributed optimization.

Submitted by Assigned_Reviewer_35

This paper proposes a randomized parallel version of ADMM, which can handle palatalization with multiple blocks. In each iteration, PADMM picks K random blocks, updates the block primal vector, and then updates the dual vector via a backward step. The authors illustrate the importance of using a backward step, as it makes the dual update more conservative, enabling global convergence. The algorithm also allows adding a proximal term to the primal update, making a part of optimization problems easier to solve. The authors give theoretical analysis on the algorithm, establishing its global convergence and its iteration complexity. When there are totally J blocks, and when the PADMM algorithm randomly picks K blocks at one time, the convergence rate of the algorithm is O(J/(TK)) after T iterations of update.

The main advantage of the PADMM algorithm is (1) it allows full parallelization for the primal step and (2) it allows relatively large update stepsize comparing to other methods. In particular, PADMM can be faster than sADMM as it allows greater stepsizes. It is more flexible than PJADMM, and more parallelizable than GSADMM. The authors have evaluated the algorithm on robust principal component analysis and
overlapping group lasso. The experimental results are quite promising. The PADMM algorithm achieves the desired accuracy with less computation time. Nevertheless, the authors haven't reported the performance of PADMM with parallel implementation. Comparing to traditional ADMM, it is less convenient to tune the three parameters of PADMM.

Overall, this is an interesting contribution to solving the ADMM-type problems, where multiple constraints are provided. The algorithm has the same convergence rate as ADMM, but it allows parallelization and it exhibits promising practical performance.
Summary: Overall, this is an interesting contribution to solving the ADMM-type problems. The paper is well written. Both the theoretical and empirical parts are solid.
Author Feedback
Author rebuttal: We thank all the reviewers for detailed and insightful comments. We will fix the typos, update the draft suitably to incorporate the feedback and address the concerns.

Reivewer16: 1. In this paper, we consider the update of all dual blocks. An extended work of PDMM shows that dual variables can be updated randomly, which will slow down the algorithm but is useful for parallel implementation.

2. Yes, the sum over i is missing in Theorem 2. Theorem 2 in the appendix has the sum over i. We will correct it.

3. PDMM has the same rate as ADMMs, so the difference lies in the constant. In Theorem 2, when choosing all blocks, i.e., K = J, the constant in PDMM is a bit smaller than splitting ADMM since PDMM considers the sparsity of A.
According to Theorem 2, the best split is the case when the constant is the smallest, i.e., J/K = 1, beta_i = 1 or d_i = K, and \tilde{L} is the smallest, which leads to tau_i =1 and K = 1. In summary, the optimal case is J=K =1, tau_i = 1, nu_i = 0, i.e., considering the entire A as a single block, which reduces to the method of multipliers. The question of how to split A needs to consider whether the resulting subproblems for each block can be solved efficiently. In robust PCA, A is split into three blocks instead of two blocks or four blocks. In Lasso, a single column should be a block since the lasso of a single column has a closed-form solution. If a block has multiple columns, extra work is required in solving the Lasso subproblem. Therefore, how to split A depends on the trade-off between the number of subproblems and how efficiently each subproblem can be solved.

4. We are not quite familiar with the tensor completion problem. In general, PDMM can randomly solve the problem with coupled equality constraints in parallel, which provides more parallelism than ADMM. The generalization of ADMM to multiple blocks with Gauss-Seidel rule still requires further study. In particular, the convergence of Gauss-Seidel ADMM and the choice of stepsize are still unclear. Compared to ADMM with splitting variables for multi-block equality constraints, PDMM can update blocks randomly and use better step size so that it can converge faster. For tensor completion problem which considers separable equality constraints, the advantage of PDMM is that it can update blocks randomly, in contrast to the update of all blocks in ADMM.

Reviewer19: In general, PDMM is slower than Gauss-Seidel ADMM without parameter tuning, as shown in the experiments of robust PCA. By tuning parameters tau and nu, PDMM is still slower than Gauss-Seidel ADMM in terms of the number of iterations (Table 1). However, Gauss-Seidel ADMM has to recompute Ax-a for each block, while PDMM can reuse Ax-a for each block. Therefore, PDMM can be faster than Gauss-Seidel ADMM in terms of runtime.

Reviewer35: The parallel performance of PDMM will be reported in an extended work. The parameters of PDMM have been given theoretically, and the computation is simple. In general, the theoretical results are good enough for PDMM to be competitive with ADMM . However, since theory only considers the worst case, tuning parameters may help PDMM converge faster.